# Identifying long-term stable refugia for relict plant species in East Asia

Cindy Q. Tang et al.[#]

Today East Asia harbors many "relict" plant species whose ranges were much larger during the Paleogene-Neogene and earlier. The ecological and climatic conditions suitable for these relict species have not been identified. Here, we map the abundance and distribution patterns of relict species, showing high abundance in the humid subtropical/warm-temperate forest regions. We further use Ecological Niche Modeling to show that these patterns align with maps of climate refugia, and we predict species' chances of persistence given the future climatic changes expected for East Asia. By 2070, potentially suitable areas with high richness of relict species will decrease, although the areas as a whole will probably expand. We identify areas in southwestern China and northern Vietnam as long-term climatically stable refugia likely to preserve ancient lineages, highlighting areas that could be prioritized for conservation of such species.

[#]A full list of authors and their affliations appears at the end of the paper.

The continent of East Asia was generally not covered by ice-sheets to the same degree as Europe and North America during the ice ages[1], enabling a greater persistence of plant taxa there (with some having originated at the Paleogene or even earlier) during the Pleistocene[2]. Accordingly, East Asia has been functioning as a region with a large number of late Neogene/Quaternary refuge areas for plant taxa[3–5]. Today, many of these relic lineages (some were relatively speciose such as Gingkoales[6]) are reduced to a few or even to a single species (Supplementary Figs. 1A–1H).

Here the broad area of East Asia is taken to include China, Japan, Korea, eastern Russia (east of ca. 80°E), Mongolia, the eastern and central Himalayas (Bhutan, Nepal, northern India, northern Myanmar), peninsular India-Sri Lanka, the deltaic plain of Bangladesh, southern Myanmar, Thailand, Vietnam, Laos and Cambodia, while Malaya, Indonesia and neighboring islands are excluded (Supplementary Figs. 2A and 3A). The climate of the area is characterized by the East Asian monsoon system (Supplementary Fig. 2A). The northwestern part of China is, apart from these monsoon systems, also influenced by the westerlies of the Northern Hemisphere[7] (Supplementary Fig. 2A), but their contribution to regional rainfall is relatively small because of their long continental trajectory. The monsoonal climate of East Asia gives rise, in combination with strong elevational diversification, to a great variety of vegetation types (Supplementary Fig. 2B). The mountains offer macroclimatic and microclimatic conditions and a complexity of habitats favoring the existence of diversified woody vegetation, ranging from tropical rain forests, to warm and humid subtropical/warm-temperate evergreen broad-leaved forests, then to temperate deciduous broad-leaved and coniferous forests, and finally to cold temperate coniferous forests of *Abies* and *Picea* (Supplementary Fig. 2B).

One of the central challenges of biology is the explanation of evolutionary processes and prediction of species richness[8,9]. Plant distributions are strongly controlled by climatic factors and changes in climate can result in the dispersal, migration, evolution/adaptation, and extinction of species[10,11]. Many plant species show a considerable degree of evolutionary and ecological conservatism: the climatic niche of a species remains unchanged over a given time period[12,13]. Ecological Niche Modeling (ENM) is widely used to predict species distributions and has been applied to identify climate refugia[14]. The term "climate refugia" has various definitions depending on the emphasis laid on the spatio-temporal scale of interest, the level of biotic organization involved, or the relationship with the core distribution range[15]. A refugium is "an area where distinct genetic lineages have persisted through a series of Tertiary or Quaternary climate fluctuations owing to special, buffering environmental characteristics"[16], or "a geographical region that a species inhabits during the period of a glacial/interglacial cycle that represents the species' maximum contraction in geographical range"[17]. Refugia can also be defined as "areas where local populations of a species can persist through periods of unfavorable regional climate"[15], and being habitats where species may survive or potentially expand under changing environments[18]. The emphasis on persistence through time is clearly reflected in the definitions of Tzedakis et al.[19]: "a location that provides suitable habitats for the long-term persistence of populations, representing a reservoir of evolutionary history", and Birks[20]: "the geographical area where particular plant and animal populations grow today, grew in the past or may persist in the future". In the present study we adopt the prerequisite of long-term persistence as the main trait defining refugia and, for the context of relict plant species in East Asia, we define "long-term stable refugia" as the climatically suitable areas that allowed the persistence (in contrast to other areas) of ancient lineages during the Pleistocene climatic oscillations and that probably will do so

under a scenario of global warming. ENM is an effective way to identify refugia, as climatic-based paleodistribution reconstructions often show good agreement with other proxies such as genetic markers[21]. In addition, ENM allows for estimating potential distribution areas, provided that future climatic models are available. Conservation priorities should be linked to areas of potential refugia that have an inherent resilience to climate change, providing safe havens where biota can be safeguarded for longest[18].

It should be noted, nevertheless, that conclusions based on ENM rely on a series of basic assumptions[22] that, if not met, may compromise in a certain degree the validity of the results found: (1) niches are conserved along time (i.e., "niche conservatism"[13]), and (2) a given species has access to all possible environmental conditions, unrestricted by barriers, dispersal disequilibrium, or negative interactions; that is, that fundamental niche can be equated to the realized niche[23]. While niche conservatism may apply in many cases[24], realized niches are generally a subset of fundamental ones[25]. Since ENM are calibrated with the observed distribution of the species, a given modeled niche actually corresponds to the realized niche and, thus, the fundamental niche could be shaped by other factors than environment (e.g., biotic interactions with other species, dispersal ability, and other abiotic factors). Despite these limitations, ENM is still a very powerful approach that is not only widely used to detect refugia, but also to discover new populations or species, to determine the impact of invasive species, to predict areas with conservation ends (for designing protected areas, or areas for restoration, translocation, or reintroductions), to evaluate the impacts of climate change on biodiversity, or to ask questions regarding the patterns of niche evolution[22].

We explore spatial distribution patterns of Paleogene-Neogene and older relict (hereinafter relict) species richness on a broad-scale, representing symbolic relict plant groups that have survived the ice ages in East Asia. We elucidate the present-day survival of relict forests from an ecological perspective and reconstruct the possible paleo-distributions of their species during the mid-Holocene and the Last Glacial Maximum (LGM) as based on ENM, to give an inclusive picture of their past distributions. Moreover, we outline potential effects of future climate change (to 2070) on relict species richness. Finally, we identify long-term (at least since the LGM to the year 2070) stable refugia maintaining ancient lineages and propose establishment of protected areas for the long-term stable refugia. We conclude that (1) relict plant species richness patterns correspond to the existence of climate refugia in East Asia, (2) favorable refugia of relict plants and forests exist in humid subtropical/warm-temperate areas of East Asia, as a result of the climatic conditions and altitudinal diversification in that area, and (3) southwestern China and northern Vietnam have provided long-term stable refugia for many Paleogene-Neogene and older relict plants.

## Results

**Present-day relict species richness patterns in East Asia.** Among the examined 133 relict genera of East Asia, 98% are woody taxa and 2% are herbaceous. There are 93 genera endemic to East Asia, and the other 40 genera have disjunct distributions (hereinafter referred to as disjunct genera) between East Asia and other parts of the world. They comprise both gymnosperms and angiosperms (Supplementary Tables 1 and 2). Of the 442 relict species in the 133 genera, 15.3% are gymnosperms and 84.7% are angiosperms. About 5.9% of the gymnosperms, and about 86.6% of the angiosperm woody species, are deciduous. Thus, among the relict species, evergreen gymnosperms and deciduous angiosperms constitute by far the largest groups, whereas deciduous

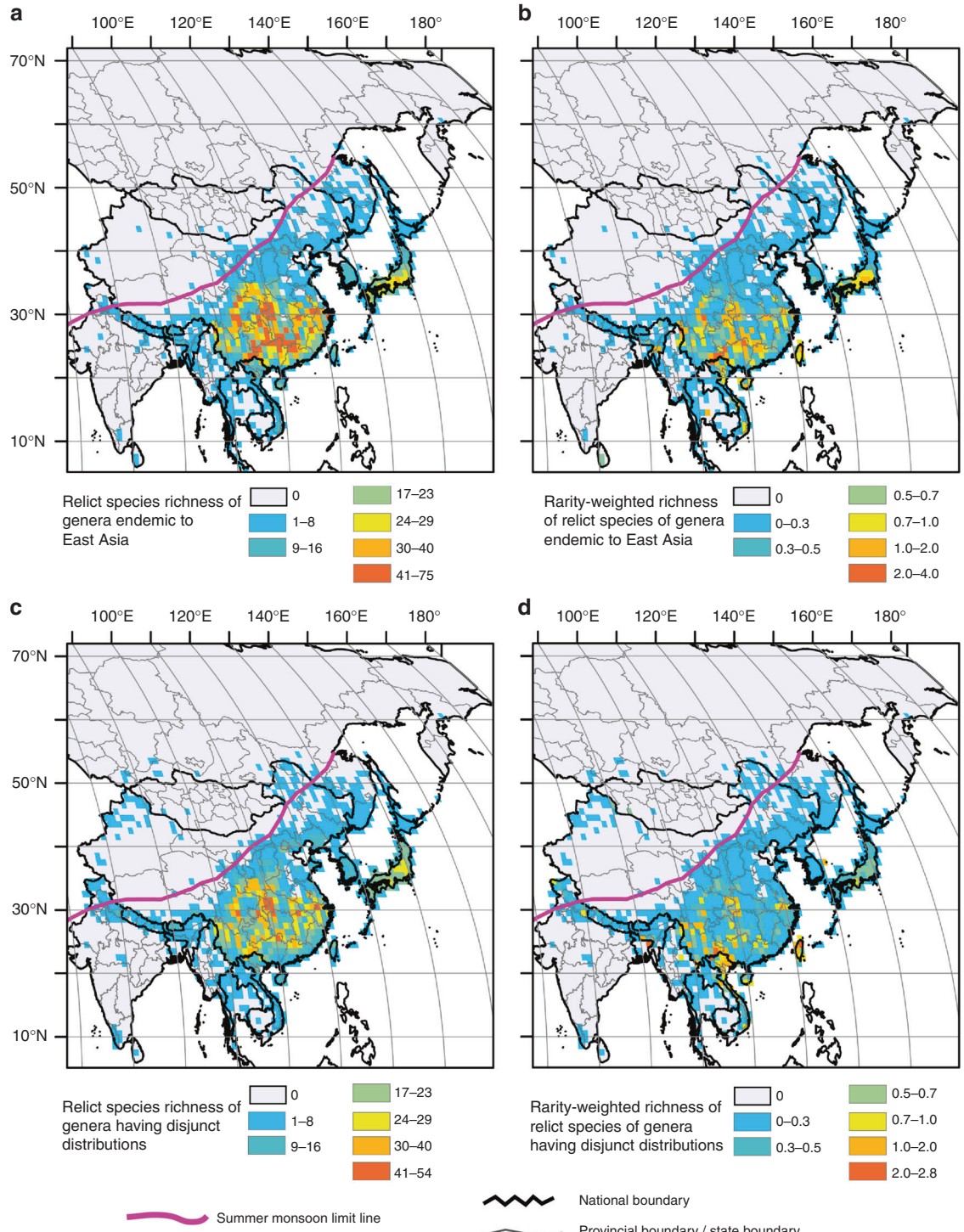

**Fig. 1** Relict species richness and rarity-weighted richness. **a** Relict species richness of genera endemic to East Asia. **b** Rarity-weighted richness of relict species of genera endemic to East Asia. **c** Relict species richness of genera having disjunct distributions between East Asia and other parts of the world. **d** Rarity-weighted richness of relict species of genera having disjunct distributions between East Asia and other parts of the world. The summer monsoon limit line adapted from Chen et al.[79], showing the approximate limit of modern summer monsoon precipitation. For the national boundary between China and India, please see Supplementary Fig. 2A and Methods. Maps were generated using the software ArcGIS v. 10.5 (ESRI, Redlands, CA, USA) and modified using Canvas 12 (ACD Systems of America, Inc., Seattle, WA, USA). Map layers were obtained from site www.gadm.org

gymnosperms and evergreen angiosperms form small groups (Supplementary Data 1 and 2).

The patterns obtained for species richness and rarity-weighted richness (the latter representing concentrations of limited-range species, as well as a high turnover of species between adjacent cells[26]) show that the relict species are mainly confined to areas to the south of the summer monsoon limit line (Fig. 1a–d), in agreement with the results of a recent study focused in China[27]. A

peculiarity is that they are represented today by many temperate woody genera characterized by a relatively high demand for humidity, as also suggested by studies on some relict plants in China[14,28]. Most of them are limited to mountains in the humid subtropical and warm-temperate areas. Notably, the mountains in the areas stretching from near the boundary between southwestern China and northern Vietnam to the subtropical/warm-temperate regions of China, then to central Japan, are characterized by high topographic heterogeneity and generally display an extremely high relict species richness and degree of endemism.

In the vast geographic region of East Asia, the distribution patterns of relict species richness (Fig. 1a, c) and rarity-weighted richness (Fig. 1b, d) have linear relationships with significant positive correlations (Pearson's $r = 0.83$, spatial autocorrelation-corrected $p < 0.0001$ using $t$-test for relict species of endemic genera; $r = 0.63$, spatial autocorrelation-corrected $p < 0.0001$ using $t$-test for relict species of disjunct genera).

The most important large-size core area with the highest paleoendemic species (species that were formerly widespread but are now restricted to a smaller area) richness (144) is situated in mountain ranges of southwestern China from the Duyang and Liuzhao Mts at the boundary between Yunnan and Guangxi, to the Miaoling Mts (Guizhou), Wumeng Mts (Yunnan), Dalou Mts (Guizhou), then to the Daliang and Longmen Mts (Sichuan), and Wu Mts (the boundary between Chongqing and Hubei), and further extends to the Daba and Qinling Mts on the boundary between Sichuan, Shaanxi and Gansu (for locations of provinces of China, see Supplementary Fig. 3B) (Fig. 1a; Supplementary Figs. 4A and 4E; Supplementary Table 3). In southwestern China the diverse topographies create a great variety of habitats that support relict species' persistence mainly at elevations of 1200–2700 m where the forests are often humid because of fogs. The five subsequent rankings of paleoendemic species richness core areas go to the mountains in south-central China, southern China, the boundaries of Yunnan, Guangxi and northern Vietnam, and southeastern China. The further subsequent rankings go to the boundary of Yunnan, Tibet and Myanmar, southwestern China (Wuliang Mts and Ailao Mts in Yunnan), and central, south-central to southern Japan (Fig. 1a; Supplementary Figs. 4A and 4E; Supplementary Table 3).

The first-rank core area of paleoendemic species rarity-weighted richness (RWR, score = 7.89) is marked by the boundary between Guangxi, Yunnan, and northern Vietnam (Daqing, Liuzhao, and Hoang Lien Son Mts). The 12 subsequent rankings of RWR are in the mountains of southwestern China, the boundary of northwestern Yunnan and Myanmar, the mountains at the rim of the western Sichuan Basin, southern, south-central, and southeastern China, Taiwan and Hainan, then central and south-central Japan (Fig. 1b; Supplementary Figs. 4B and 4E; Supplementary Table 4).

The major core areas of relict species richness of disjunct genera are, beginning with the most important, southwestern, southern and south-central China, the boundary between Yunnan and Guangxi and northern Vietnam, southeastern China, then to the boundary between Yunnan, Tibet and Myanmar, finally to south-central and central Japan (Fig. 1c; Supplementary Figs. 4C and 4E; Supplementary Table 5). The most notable core area of RWR (11.50) of disjunct taxa on the boundary of Guangxi, Yunnan and northern Vietnam shows a spatial pattern similar to the RWR of paleoendemic taxa. The next is south-central and southwestern China, followed by the southeastern Himalayas, the boundary of Guizhou and Guangxi in southwestern China, Taiwan, northern Vietnam, central Laos, the boundary of Yunnan and Guangxi in southwestern China, southern China, southeastern China, Hainan Island, and

south-central Japan (Fig. 1d; Supplementary Figs. 4D and 4E; Supplementary Table 6).

While most of the relict species are confined to mountains in the humid subtropical and warm-temperate areas at the present, the relict genera, both the endemic and those having disjunct distributions of fossil records, evidently had broad distribution ranges in the Northern Hemisphere during the Paleogene and the Neogene Periods, and some of them even in the Cretaceous (Supplementary Note 1, Supplementary Figs. 5 and 6).

**Present-day relict forests in East Asia.** The mountains in the humid subtropical/warm-temperate areas of East Asia are home to extraordinarily abundant relict forests. About 422 forest stands in 188 forest types dominated by relict species thrive between the tropical and temperate forest zones in the mid-latitudes (22°–37°), mostly from the boundary between southwestern China and northern Vietnam to the subtropical/warm-temperate regions of China, then to central Japan, mainly at altitudes of 1500–2700 m within latitudes 22°–25° N, at 800–2000 m within 25°–32° N, and at 500–1500 m within 32°–37° N (Fig. 2a–d).

There are 75, 102, and 11 types of natural forest stands, dominated respectively by relict gymnosperms, relict deciduous broad-leaved woody species, and relict evergreen broad-leaved tree species (Supplementary Data 3). Many of those forests thrive in complex habitat mosaics of mountain topography where mountain ranges facilitate orographic precipitation, as exemplified by forest stands dominated by paleoendemic and monotypic *Ginkgo*, *Metasequoia*, *Taiwania*, *Cathaya*, *Cryptomeria*, *Glyptostrobus*, *Pseudolarix*, *Sciadopitys*, *Thujopsis*, *Davidia*, *Tetracentron*, *Cercidiphyllum* and *Rhoiptelea*. Some stands dominated by oligotypic genera (genera of plants that contain only a few species) having disjunct distributions include *Thuja*, *Pseudotsuga*, *Taxus*, *Liriodendron* and *Nyssa* are also good examples. A great number of these stands are distributed in the altitudinal transition zone from the evergreen broad-leaved to the deciduous broad-leaved forests or from the broad-leaved to the coniferous forests. Among the 422 relict forests, 92% of the stands are found in mountain stream flood plains and ravines, and on scree or steep slopes, cliffs and rocky terrains or limestone areas where natural disturbances often occur. Thus, as a general rule, relict species are restricted to local habitats with little competition with non-relict species in the regeneration phase, and most of them appear to be shade-intolerant and to belong to the stress-tolerant type[29]. Relict plants are usually pioneer or seral species and, often, at least one of the relict species is a dominant in early or seral successional forest associations, with their persistence seemingly favored by natural disturbances[30–34].

**Relict species richness patterns shaped by climate changes.** All ecological niche models (see Methods section) performed well with both jackknife (with a mean probability of success = 0.846) and continuous Boyce index (mean CBI = 0.787) approaches, suggesting a high fit of the model. Based on the number of species in which a given variable was selected, but also on its percent contribution to the individual models, the most important bioclimatic variable controlling the geographical distribution of these relict species as a whole was the mean temperature of the coldest quarter (bio11), followed by isothermality (bio3) and precipitation seasonality (bio15).

The predicted suitable areas for the high relict species richness (ranks of 51–128 and 21–50) of both endemic and disjunct genera under the present climate are mainly located in the subtropical and warm-temperate regions of China, northern Vietnam and Japan, as well as in the boundary area between northwestern Yunnan and Myanmar. These distributions are generally

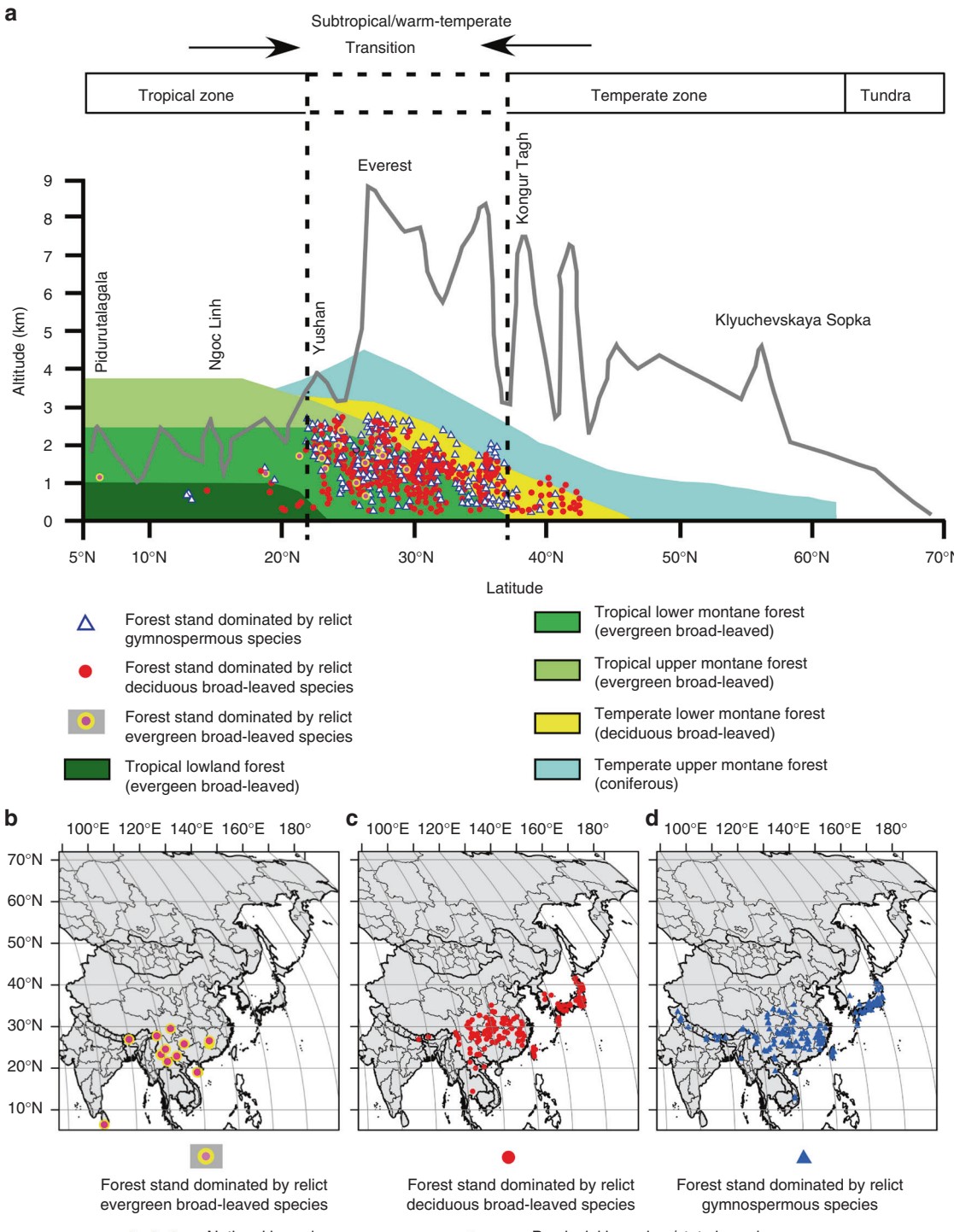

**Fig. 2** Distribution of relict forest stands in East Asia. **a** Relict forest stands overlaid on a potential forest vegetation zone model and mountain profile of East Asia. The potential forest vegetation zone model is modified from Ohsawa[80]. **b–d** Latitudinal and longitudinal distribution of relict forest stands in East Asia. For the national boundary between China and India, please see Supplementary Fig. 2A and Methods. Maps were generated using the software ArcGIS v. 10.5 (ESRI, Redlands, CA, USA) and modified using Canvas 12 (ACD Systems of America, Inc., Seattle, WA, USA). Map layers were obtained from site www.gadm.org

consistent with the observed present relict species richness patterns (Figs. 1a, c and 3a, h). The areas in the Himalayas and the boundary between northwestern India and Myanmar are predicted to be suitable for a rather high relict species richness, based on climate variables; however, high richness is not found

there at the present time (Figs. 1a, c and 3a, d). Reasons for this discrepancy may include the geologic youth of these areas that were tectonically very active during the last 10 million years[35,36]; a well-known prerequisite for relict species persistence is tectonic stability[37,38], while dispersal limitations that are usually

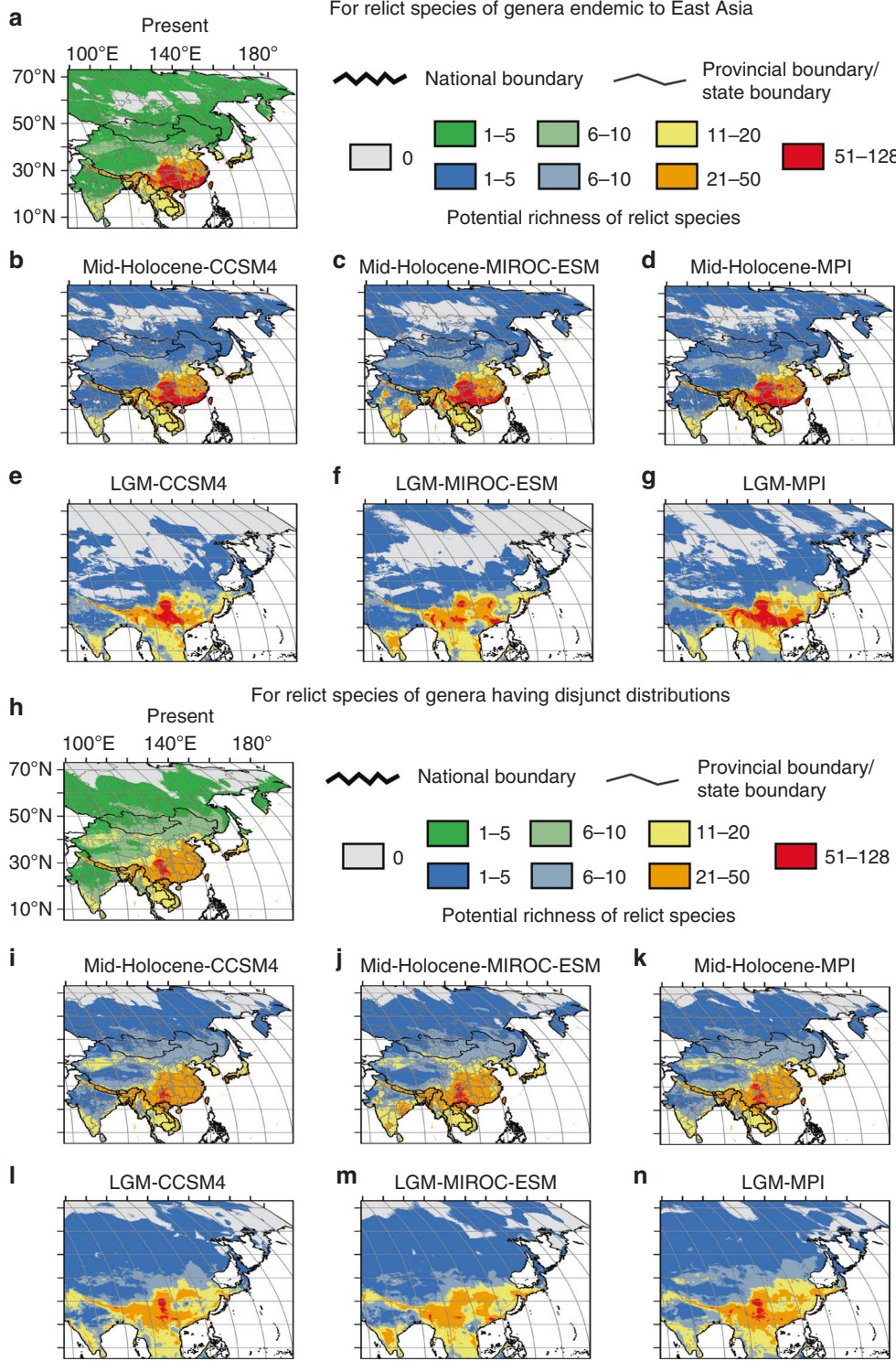

**Fig. 3** A comparison of potential richness of relict species under the present climate and climatic scenarios in the past. **a–g** For potential richness of relict species of genera endemic to East Asia: **a** under the present climate; **b–d** under scenarios mid-Holocene-CCSM4, mid-Holocene-MIROC-ESM, mid-Holocene-MPI, respectively; **e–g** under scenarios LGM-CCSM4, LGM-MIROC-ESM, LGM-MPI, respectively. **h–n** For potential richness of relict species of genera having disjunct distributions: **h** under the present climate; **i–k** under scenarios mid-Holocene-CCSM4, mid-Holocene-MIROC-ESM, mid-Holocene-MPI, respectively; **l–n** under scenarios LGM-CCSM4, LGM-MIROC-ESM, LGM-MPI, respectively. For the national boundary between China and India, please see Supplementary Fig. 2A and Methods. Maps were generated using the software ArcGIS v. 10.5 (ESRI, Redlands, CA, USA) and modified using Canvas 12 (ACD Systems of America, Inc., Seattle, WA, USA). Map layers were obtained from site www.gadm.org

associated with many relict species would have avoided the arrival of ancient lineages in these areas.

As compared to the present, in the mid-Holocene (ca. 6000 year BP) under three scenarios (Methods), the predicted suitable areas as a whole were slightly reduced (by 4.7% on average) for relict species of endemic taxa, and they increased by 2.9% on average for relict species of disjunct taxa. In the LGM (ca. 21,000 year BP) under three scenarios (Methods), the areas as a whole decreased by 11.6% and increased by 39.7% on average in relict species of endemic and disjunct taxa, respectively (Supplementary Tables 7 and 8). The predicted suitable areas, especially the core areas with the highest relict species richness (51–128) fell mostly within the current potential distribution ranges of both endemic and disjunct genera except for disjunct taxa under the scenario LGM-MIROC-ESM, though the areas decreased by somewhat (20.4% on average) (Fig. 3a–n, Supplementary Tables 7 and 8).

As compared to the present, in the future (2070) under six scenarios (Methods), the predicted suitable core areas with the highest richness (51–128) of relict species will decrease (27.8% and 68.1% on average for relict species of endemic and disjunct genera, respectively), which affects mainly their southern ends. Suitable areas with moderate richness (11–50) also show slight contractions in southern parts of the study region. Such areas of moderate richness would also be generally more or less expanded northward (e.g., northeastern China, North Korea, northern Japan, and southern Kamchatka of the Russian Far East) for both endemic and disjunct genera (Fig. 4a–n; Supplementary Tables 7 and 8), though the northward expansion would be small. However, those habitat gains are unlikely to be accessible to the relict plants because of dispersal limitations of these species, and the sustainability of its habitat will not be maintained unless conservation strategies are introduced. In contrast, current climatically suitable areas in southern and central Japan will strengthen, and new potential areas will form in the southern Korean peninsula.

**Long-term stable refugia**. Figure 5a–j show the overlap of the same climatic space for relict species persisting across time to reveal areas that remain stable in climatic variables.

The overlap of the past (the LGM and the mid-Holocene) and the present (Methods) shows that mountain areas (518,100 km² in size), mainly in southwestern China (the Yunnan-Guizhou Plateau, the Sichuan Basin), northernmost Vietnam, and the boundary between Guangxi, Guangdong, and Hunan in southern China, have been suitable climate refugia with high relict species richness of endemic genera (26–85 species occupying the same climatic space from the LGM to the present) (Fig. 5a; Supplementary Table 9). Meanwhile the overlap areas (442,250 km² in size), mainly from northern Vietnam, and Yunnan to the Sichuan basin, have maintained a high relict species richness (26–85) of disjunct genera (Fig. 5f; Supplementary Table 10) since the LGM.

Compared to the overlap of the past (the LGM and the mid-Holocene) and the present, the overlap of the present and the future (2070) (Methods) shows that potential overlap areas with high relict species richness (26–85) of endemic and disjunct genera would be larger (1,128,270 km² on average for endemic genera and 604,990 km² on average for disjunct genera) in subtropical China (Fig. 5b, c, g, h; Supplementary Tables 9 and 10).

The overlap of the past (the LGM and the mid-Holocene), the present and the future (2070) (see Methods section) shows that mountain areas (237,910 km² in size) mainly from northern Vietnam, southeastern and eastern Yunnan, to western and northern Guizhou, then to the Sichuan Basin of southwestern

China, and some very small additional areas in west-central Yunnan and around the boundary between Guangxi, Guangdong and Hunan, are predicted to be long-term stable areas with a high relict species richness (26–85) of endemic genera, while overlap areas (114,890 km² in size) of high relict species richness (26-85) among disjunct genera are mainly located in northern Vietnam, southeastern, eastern and northern Yunnan, western Guizhou and the boundary between Sichuan and Chongqing under the scenarios of Present-Past-Future RCP 2.6 (Fig. 5d, i; Supplementary Tables 9 and 10). Under scenarios of Present-Past-Future RCP 8.5, the overlap areas (52,530 km² in size) with high relict species richness (26–85) of endemic genera are very much reduced as a whole, while for the relict species of disjunct genera overlap areas of just 7460 km² in size are located only in southeastern Yunnan, around the boundary with northern Vietnam, and the Hoang Lien Son mountain range (Fig. 5e, j; Supplementary Tables 9 and 10).

This reveals that, given climatic stability, many mountains in southwestern China (Yunnan-Guizhou Plateaus including southeastern, eastern and northern Yunnan, western and northern Guizhou, the Sichuan Basin including eastern Sichuan and Chongqing) and northern Vietnam can be regarded as long-term stable refugia for relict species.

The relationships between the present-day relict species richness and the number of species occupying the same climatic space from the LGM to the present have significant positive linear correlations (Pearson's $r = 0.75$, $p < 0.0001$ using $t$-test for species of endemic genera; $r = 0.6$, $p < 0.0001$ using $t$-test for species of disjunct genera). Thus, the existence of climate refugia (from the LGM to the present) might largely explain the present-day richness patterns of relict species in East Asia. Our results highlight that on a broad geographic scale, the survival and richness patterns of relict species and their forests in East Asia have been mainly shaped by climate.

## Discussion
The identified long-term stable refugia (the areas in red in Fig. 6a) have been largely determined by the occurrence of mild temperatures during the coldest quarter throughout time (LGM to year 2070), as bio11 clearly increases (but its variability decreases) for the areas where the number of climatically suitable species is higher (Fig. 6b). In other words, the mean temperatures of the coldest months within the long-term stable refugia have rarely gone below 0 °C for any time period, including the LGM, while in other regions within the study area, temperatures could go much lower (even reaching −30 °C, Fig. 6b). This is not an unexpected result as relict species were formed in a time when the world had much higher temperatures as compared to the Quaternary (with differences that would have exceeded 10 °C[39]; fossils of relict lineages were left at Arctic latitudes until the Pliocene (Supplementary Fig. 5). Another climatic prerequisite of the long-term stable refugia is a low level of isothermality (bio3), which is defined as the ratio of the temperature mean diurnal range (bio2) to the temperature annual range (bio7); as seen in Fig. 6c, refugia are places where the day-to-night temperature range has rarely surpassed 50% of the summer-to-winter range. In contrast, surrounding areas might reach values of 70% (i.e., nearly an absence of seasonal variation in energy). Finally, the long-term stable refugia are areas with a relatively high precipitation seasonality (with a coefficient of variation generally ranging from 60 to 100%) compared to surrounding areas, where these percentages reach substantially lower or higher values (Fig. 6d). These low and high values for bio3 and bio15 are somewhat surprising, as global tree species richness is generally associated with high isothermality and low precipitation seasonality[40]. A possible

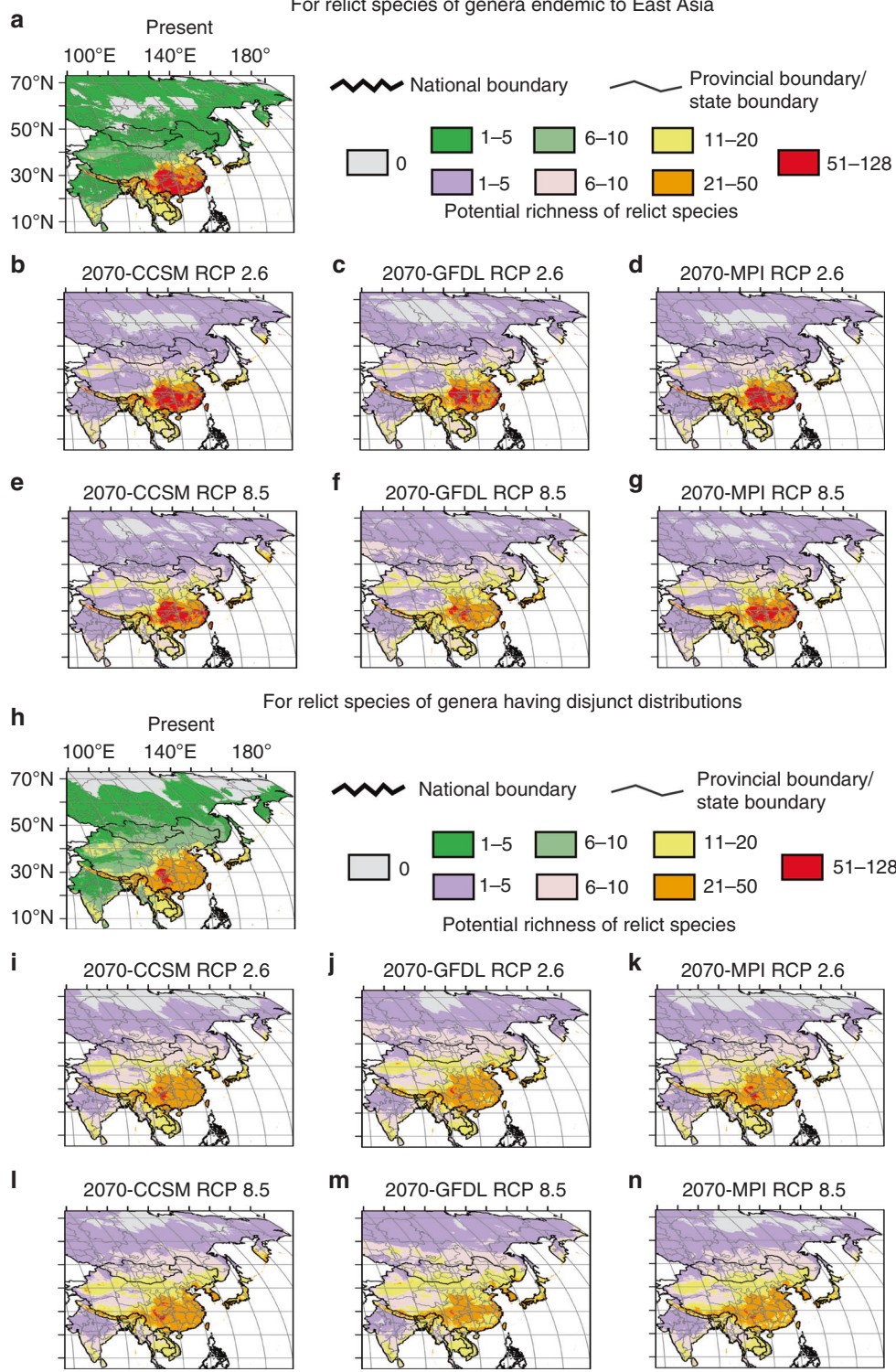

**Fig. 4** A comparison of potential richness of relict species under the present climate and climatic scenarios in the future (2070). **a–g** For potential richness of relict species of genera endemic to East Asia: **a** under the present climate; **b–d** under scenarios 2070-CCSM RCP 2.6, 2070-GFDL RCP 2.6, 2070-MPI RCP 2.6, respectively; **e–g** under scenario 2070-CCSM RCP 8.5, 2070-GFDL RCP 8.5, 2070-MPI RCP 8.5, respectively; **h–n** For potential richness of relict species of genera having disjunct distributions: **h** under the present climate; **i–k** under scenarios under scenarios 2070-CCSM RCP 2.6, 2070-GFDL RCP 2.6, 2070-MPI RCP 2.6, respectively; **l–n** under scenarios 2070-CCSM RCP 8.5, 2070-GFDL RCP 8.5, 2070-MPI RCP 8.5, respectively. For the national boundary between China and India, please see Supplementary Fig. 2A and Methods. Maps were generated using the software ArcGIS v. 10.5 (ESRI, Redlands, CA, USA) and modified using Canvas 12 (ACD Systems of America, Inc., Seattle, WA, USA). Map layers were obtained from site www.gadm.org

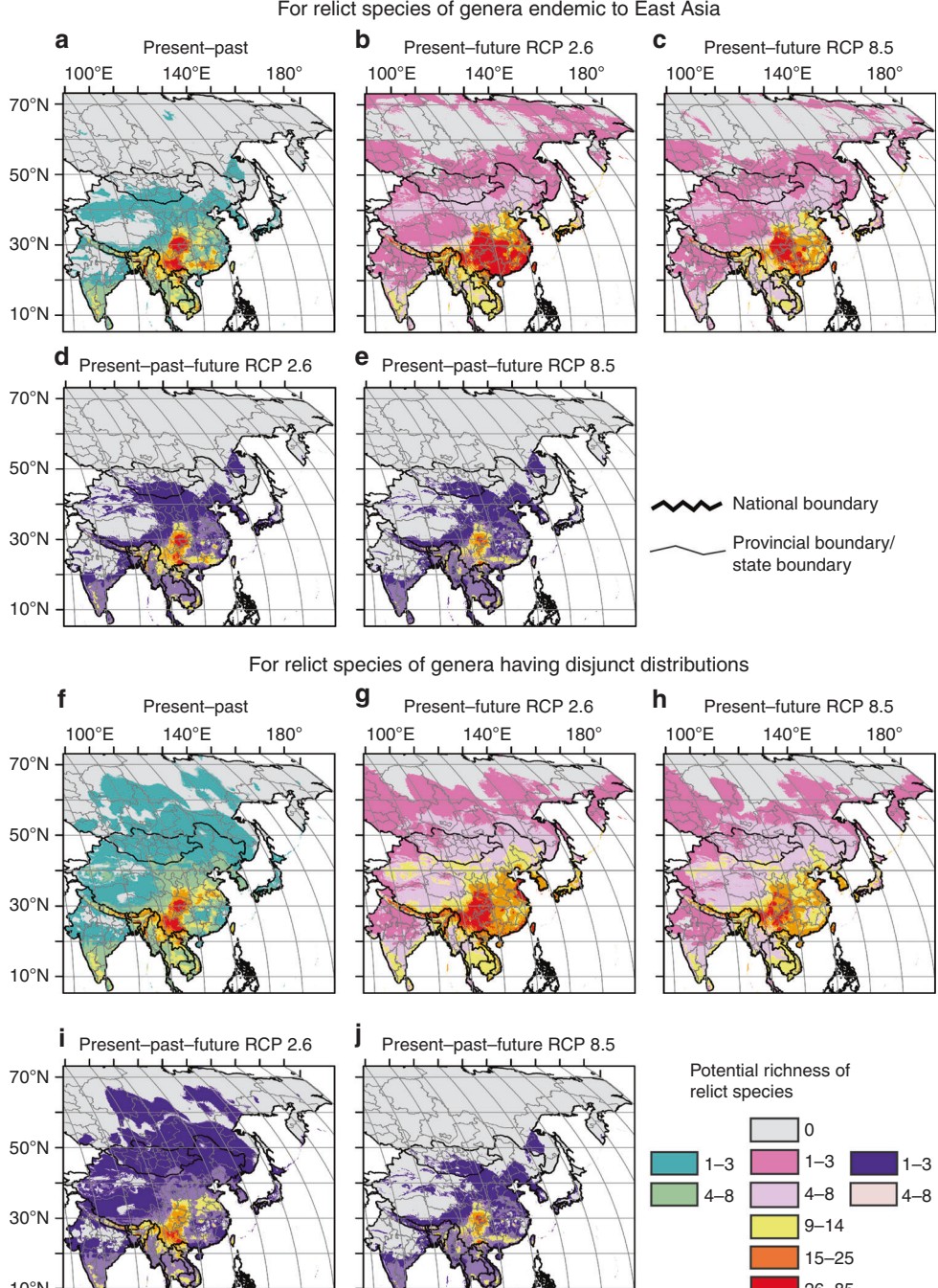

**Fig. 5** Overlap of potential richness of relict species of the four time frames climatic scenarios. **a–e** For relict species of genera endemic to East Asia: **a** Overlap of the present and the past (LGM, mid-Holocene); **b**, **c** Overlap of the present and the future (2070) (under scenarios of RCP 2.6 for **b**, RCP 8.5 for **c**); **d**, **e** Overlap of the present, the past and the future (under scenarios of RCP 2.6 for **d**, RCP 8.5 for **e**). **f–j** For relict species of genera having disjunct distributions: **f** Overlap of the present and the past; **g**, **h** Overlap of the present and the future (under scenarios of RCP 2.6 for **g**, RCP 8.5 for **h**); **i**, **j** Overlap of the present, the past and the future (under scenarios of RCP 2.6 for **i**, RCP 8.5 for **j**). For the national boundary between China and India, please see Supplementary Fig. 2A and Methods. Maps were generated using the software ArcGIS v. 10.5 (ESRI, Redlands, CA, USA) and modified using Canvas 12 (ACD Systems of America, Inc., Seattle, WA, USA). Map layers were obtained from site www.gadm.org

explanation is the progressive shift towards more seasonal climates throughout the Oligocene and Miocene[39,41], which was especially evident with the onset of the East Asian monsoons at late Miocene[42,43], to which the (relict) plants had to adapt. In sum, only some mountainous areas of southwestern China and northern Vietnam have allowed the persistence of a large

ensemble of relict plants throughout the Quaternary (LGM-2070), because they are characterized by mild climates likely resembling those of the Miocene; such areas have (1) warm climates (with annual mean temperatures between 10 and 20 °C) that are frost-free even during the winter, (2) have humid climates (with annual precipitation between 1000 and 2000 mm),

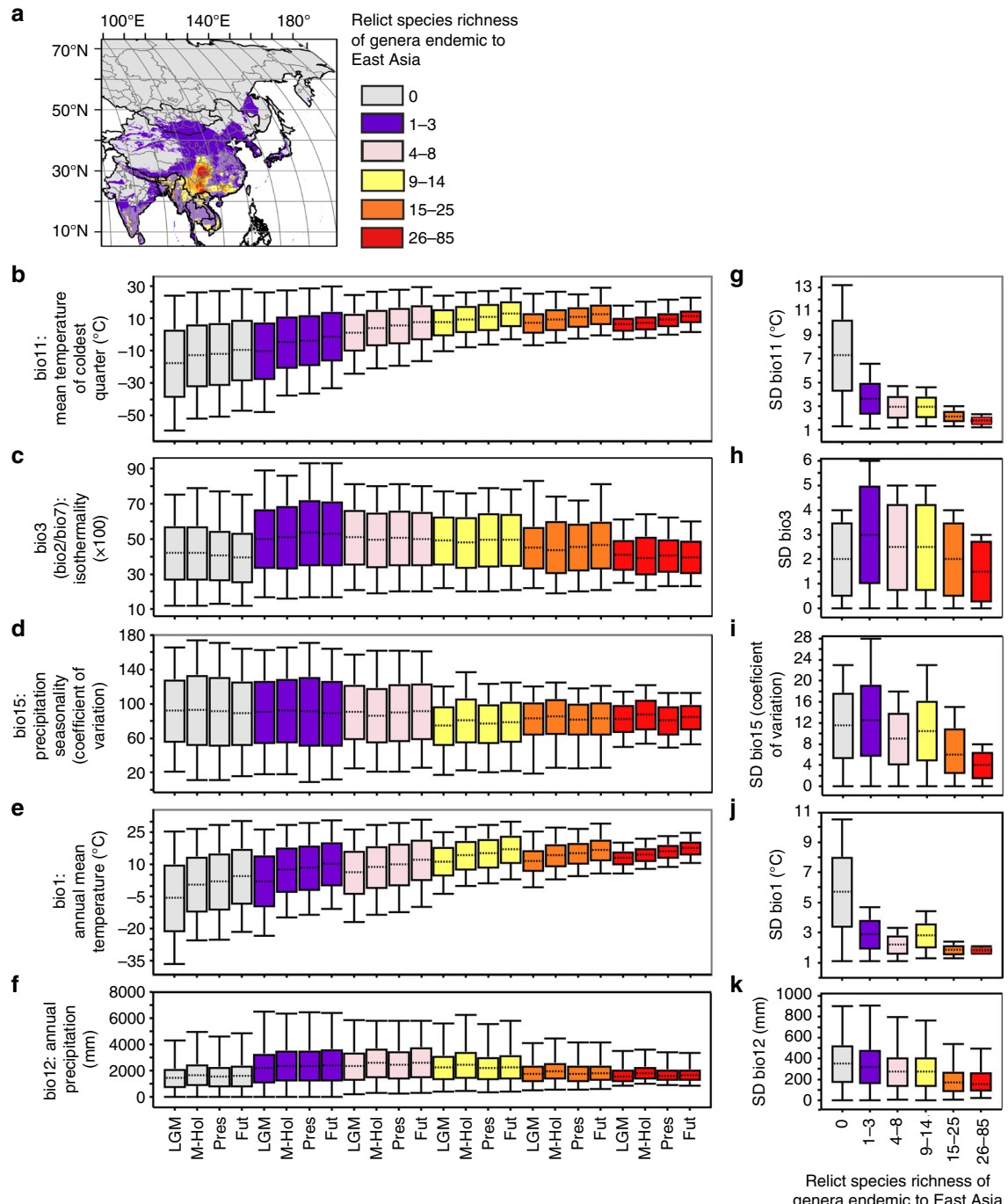

**Fig. 6** Climatic variables in different categories of suitable areas of relict species. **a** Long-term stable refugia (red areas) of relict species of genera endemic to East Asia under scenarios Present-Past-Future RCP 2.6. Maps were generated using the software ArcGIS v. 10.5 (ESRI, Redlands, CA, USA) and modified using Canvas 12 (ACD Systems of America, Inc., Seattle, WA, USA). Map layers were obtained from site www.gadm.org. **b–f** Five climatic variables in different categories of suitable areas of relict species of genera endemic to East Asia in four time frames. **g–k** Value range of the standard deviation (SD) of each climatic variable in different categories of suitable areas. LGM, Last glacial maximum; M-Hol, Mid-Holocene; Pres, Present; Fut, Future (2070). **b–k** Boxplot representation: center line (median), upper end line whisker (maximum value), upper box bound (75th percentile), lower box bound (25th percentile), lower end line whisker (minimum value)

and (3) show a moderate seasonality both in temperature and precipitation (Fig. 6b–f). Paleoecological records indicate that warm, humid and seasonal climates would have been common in southwestern China at least from the middle-late Miocene[44–47], which emphasizes, or perhaps underlines, the long-term stability of climatic conditions as a sine qua non for relict species survival.

In contrast, the absence of such favorable climatic conditions for the thermophilic, moisture-loving, and mesophytic relict lineages throughout time in other parts of East Asia, but also in Europe and North America, caused their local extinction[5,48–50].

Topographically diverse landscapes can buffer regional climate variability, creating stable climatic conditions for plant

species[51,52]. In this way the mountains of southwestern China and northern Vietnam served as suitable refugia for relict species at least since the LGM (and probably since the beginning of the Pleistocene, when a cold but especially dry climate spread throughout the whole area[42], making many habitats unsuitable for thermophilic and humidity-dependent species). Accordingly, we identify the mountains of southwestern China and northern Vietnam as long-term stable refugia from the Pleistocene to the present and future. The identified long-term stable refugia (i.e., the areas in red in Fig. 6a), in addition to being characterized by mild and moderately-seasonal climates, remained markedly stable compared to surrounding areas; indeed, the lower potential number of relict species for a given area, the larger the standard deviation for any bioclimatic variable (Fig. 6g–k). Our results, therefore, clearly support the view that climatic stability is a prerequisite for plant species refugia[18,51,53,54]. Within the refugia that result from the complex habitat mosaics of mountainous topography, a great number of relict plants appearing as pioneer and seral species are restricted to local habitats with moderate disturbance regimes where competition usually is less intense and the regeneration potential of non-relict species is lower (Supplementary Fig. 7). Climatically stable refugia harbor not only high concentrations of relict plants, but also greater concentrations of endemic species, as well as high species diversity[38,55]; indeed, long-term stable climate refugia are also long-term stores of genetic diversity, and thus might also be foci of speciation[11,52,55]. The long-term stable refugia of southwestern China and northern Vietnam conserve ancient lineages, in many cases lineages that became extinct before and during the LGM in North America and Europe, but persisted to the present in East Asia. Highest priority for conservation should be given to the long-term stable refugia.

Establishment and management of protected areas (nature reserves) must include recognition of the identified refugia. In these long-term stable refugia an average of up to 80.1% of the area with relict species of endemic genera and, on average, up to 73.1% of the area with relict species of disjunct genera, are outside the existing nature reserves of southwestern China and northern Vietnam (Supplementary Figs. 8A–8E; Supplementary Table 11). To maintain the current genetic diversity in the populations of these relict species, as seen in Supplementary Figs. 8A–8E, the most urgent need for new nature reserves is located in the mountains of the Sichuan Basin and in southeastern, eastern and northern Yunnan, as well as the northernmost provinces and the Hoang Lien Son mountain range of Vietnam.

## Methods

**Specific definitions, terms, and notations**. To define taxa that have survived the ice ages, we deliberately restrict our choice to relict genera of Paleogene-Neogene, or earlier than the Paleogene, origin with mega/macrofossil (seeds, seed cons, fruits, foliage, or woods) records dating from before the Pleistocene, or phylogenetic evidences that meet one of two criteria: (1) genera endemic to East Asia (but genera that extend further and are predominantly distributed in Malaysia and neighboring islands are beyond the scope of this paper); (2) oligotypic (genera of plants that contain only a few species) or small genera exhibiting intercontinental or European-East Asian or West Asian-East Asian disjunct distribution patterns. The criteria allow us to exclude possible neoendemic species of any large genus while providing confidence that all species of the genera included are relict taxa to the best of our knowledge. Taking an ecological and evolutionary point of view, we excluded infraspecific taxa from our list of selected relicts. In total, 133 genera are selected (Supplementary Tables 1 and 2 ). Among them, 63 paleoendemic genera, and 36 relict genera with disjunct distributions between East Asia and other parts of the world, have been confirmed by mega/macrofossil records; 30 paleoendemic genera and 4 relict genera with disjunct distribution without fossil records, have been confirmed by primitive morphology or published phylogenetic studies.

On maps in this paper borders between countries are drawn in continental areas only. All co-authors of this joint paper declared themselves respectfully concordant with the official statement on borders between countries, as declared by their own governments. We have marked the boundary lines between China and India (at issue) on the maps in Supplementary Fig. 2A, Supplementary Figs. 3A and 3B. On other maps, the (contested) boundary line showing Arunachal Pradesh to be in

India is used, since the data on relict species distribution in Arunachal Pradesh were made available by an Indian administrative organization.

**Data collection and analysis**. After making a list of East Asian relict genera of plants according to our criteria, we compiled the largest existing collection of present-day species occurrences of the genera that have survived the ice ages. The locality data for species were based on digitization of local herbarium specimen labels from physical herbaria, in addition to plant distribution databases and field records from our own fieldwork during recent decades. Further data were drawn from a large number of printed sources including e-floras, monographs and journal articles pertinent to the flora of East Asia. A list of herbaria, websites and literature for our data collection is provided in Supplementary Data 4. Data were scrutinized for misidentified, possibly inaccurate or duplicate records by the expertise and further field work of botanists and plant ecologists who have contributed to this paper. We georeferenced localities based on the descriptions of the sites where species grow; we considered only those descriptions that allowed georeferencing with the accuracy required for ecological niche modeling (2.5 arc-min, ~5 km). For species that favor human cultivation, we only recorded data on their natural occurrences as based on our own expertise and field work. If there was an uncertainty regarding cultivation status on some sites, we do not include the sites in our species occurrence dataset so as make the distribution data of the relict species as accurate as possible. For the number of occurrences for each species please see Supplementary Data 1 and 2.

We respect the usage of accepted species names in updated flora books from each country in East Asia. But when several names are used for a single species in various countries, we selected the accepted names following http://www.theplantlist.org/1/ and http://www.catalogueoflife.org/ (retrieved data during May 2016–January 2017).

We acquired data from vast literature that covers all the sites of mega/macrofossil records of our selected genera and georeferenced the fossil localities to illustrate the historical distributions. The major sources documenting these fossil records are published journal articles, monographs and the paleobiological database (www.paleobidb.org, retrieved data during March 2016–March 2017). If a locality in a review paper was not clear, we checked the original paper to clarify the occurrence. In total, after removing duplicate records of a genus at same occurrence locality, we obtained 3192 records of fossil localities for 99 out of 133 paleo-genera treated in this paper.

Species richness (number of species) and the rarity-weighted richness (RWR) (each species is assigned a score—based on the inverse of the number of sites in which a species occurs)[56] are used to show the present relict species distribution patterns. A square of 1° latitude × 1° longitude (a cell/a grid size) was chosen as a geographic unit to show species richness and rarity-weighted richness at present on the map using ArcGIS v. 10.5. The correlation of species richness and rarity-weighted richness was analyzed using Pearson's correlation coefficient. The related $p$-value was corrected for the spatial autocorrelation with Dutilleul's[57] method and implemented with R Package SpatialPack v. 0.3[58] (http://spatialpack.mat.utfsm.cl). Among cells, we consider, the top 20%, those having species richness of at least two consecutive cells, to be the core areas of relict species richness. RWR can efficiently represent rare species in a given area[59]. We consider polygons, composed of at least two consecutive cells, each with a score of at least 0.7 RWR, as the major areas of rare relict species.

To synthesize local habitats of relict plants and forests, we conducted fieldwork and literature analysis.

**Ecological niche modeling**. Species distribution models were built only for those species with at least five occurrence data (yielding 220 relict species of genera endemic to East Asia, and 163 of genera having disjunct distributions), given that the predictive ability of models with sample sizes below five may not be enough[60]. We compiled as baseline predictors a set of 19 WorldClim bioclimatic variables[61], which were extracted for current climatic conditions (ca. 1960–1990), and cover the study area (0–80°N and 65–180°E). From the WorldClim database (www.worldclim.org) we also downloaded bioclimatic variables for three time slices: mid-Holocene (MH, ca. 6000 year BP), LGM (ca. 21,000 year BP) and 2070 (average for 2061–2080). For the past (the mid-Holocene and LGM), we employed data derived from three general circulation models (GCMs) that are available in WorldClim: the Community Climate System Model Version 4 (CCSM4)[62], the Model for Inter-disciplinary Research on Climate Earth System Model (MIROC-ESM)[63], and the New Earth System Model of the Max Planck Institute for Meteorology (MPI-ESM-P)[64]. For the year 2070, we used three of the models that have shown excellent performance among those that have participated in the 5th Coupled Model Inter-Comparison Project (CMIP5) experiment[65]: CCSM4, the NOAA Geophysical Fluid Dynamics Laboratory Coupled Model 3 (GFDL-CM3)[66], and MPI-ESM-LR. The three models were run in two of the four representative concentration pathways (RCPs) of the Fifth Assessment IPCC report, RCP 2.6 and RCP 8.5[67]. Whereas RCP 2.6 represents the most "benign" scenario (i.e., a likely increase of 0.3–1.7 °C for ca. 2081–2100), RCP 8.5 is the most extreme one (a likely increase of 2.6–4.8 °C for ca. 2081–2100). All 19 bioclimatic variables for present, past and future climate scenarios were downloaded with a resolution of 2.5 arc-min (~5 km).

To select the species-specific environmental set from 19 variables, we selected a set of candidate models after correlation analysis for environmental variables in

5000 random points generated from background bias file[68]. First, we excluded those models that included combinations of highly correlated variables (Pearson's $r \geq |0.70|$; correlation analyses were run in R package base[69]). Then, we calculated Variance Inflation Factors (VIF); datasets with VIF $\geq 5$ were excluded to avoid multi-collinearity. VIF were estimated using the vif function included in the usdm package in R[69]. By this procedure, the number of final candidate combinations of parameters was 2811. To select the most parsimonious combination of parameters for each species, we first set the ß-multiplier at 0, and chose the best combination among 2811 candidate parameter combinations by the corrected Akaike Information Criterion (AICc)[70]. Then, we tested 31 different ß-multipliers[70] from 0 to 15 in steps of 0.5 and chose the best ß-multiplier value based on AICc.

Having chosen the best combination of variables for each species, we employed the maximum entropy algorithm as implemented in MaxEnt v. 3.3[71] to get distribution models at present, which were further projected to the past (mid-Holocene and LGM) and to the future (2070). As input features, we selected only "linear" and "quadratic" to avoid producing excessively complex models, which could lead to extrapolation errors[72]. As spatial bias in species distribution data is a general phenomenon in distributional databases due to the uneven sampling effort, we used the "background manipulation with bias files for a target group of species" method, which approximates the sampling distribution by combining occurrence records for a target group of species that are all collected or observed using the same methods when the sampling distribution is not known[68,73]. We used two methods of replication to construct MaxEnt models: jackknife for those species with 5–29 occurrence records (which is the recommended method for species with low sample sizes[60]), and cross-validation for those with more than 30 occurrences. For the cross-validation approach, we ran 10 replicates to obtain more robust modeling results. For the jackknife approach, we ran replicates with the same number of occurrence points. Predictive performance of models was tested with a jackknife (or "leave-one-out") procedure[60], through success rate ($q$, which is the proportion of right predictions) and statistical significance (a $p$-value computed across the set of jackknife predictions); additionally, for those species with more than 30 occurrences, model performance was assessed using the continuous Boyce index (CBI[74,75]). To distinguish between suitable and not suitable areas in the models, we chose the maximum sensitivity plus specificity (MSS) logistic threshold, which is very robust with all types of data[76]. Finally, to overcome the uncertainty associated to the variability in the predicted suitable areas among all generated models, we employed the methodology described in the ref. [14]. Following this method, once the "standard" models (those averaging all the runs, either obtained from the cross-replication or the jackknife) for each species were obtained, we removed the pixels that were regarded as suitable by <95% of the replicate models; the resulting maps were regarded as "refined" maps (i.e., those depicting predicted areas with a very high level of confidence[14].

Once "refined" maps for each species were obtained, we converted these into binary or presence/absence maps (applying the MSS threshold), which were then stacked (i.e., summed[77]) to generate maps of species richness. These maps, drawn with the help of ArcGIS v. 10.5 (ESRI, Redlands, CA, USA), were created separately for relict species of genera endemic to East Asia (that is, on the basis of 220 individual ENMs) and for those of genera having disjunct distributions (163 ENMs) for all time periods (present, mid-Holocene, LGM, year 2070) and using all GCMs. With this method, we are able to detect centers of species richness, but these centers are just delineated on the basis of detecting pixels of species accumulation, irrespective of what these species are; in other words, a given center of richness can be maintained through time (e.g., from LGM to 2070), but the species accumulating there can be different. Identifying areas of richness with the same species persisting across time, in contrast, is indicative of the occurrence of long-term stable refugia; these areas should merit the highest priority for conservation[78]. To achieve this, individual species binary maps at different time scenarios were intersected, with only the suitable areas that overlap across time being retained; such maps were again stacked (separately for relict species of genera endemic to East Asia and for relict species of genera having disjunct distributions) to generate maps of long-term stable refugia. We assessed five different types of time combinations (Supplementary Fig. 9): (1) present with all past models (expressed as Present–Past in the text and Figures); (2) present with all future models of RCP 2.6 (expressed as Present–Future RCP 2.6); (3) present with all future models of RCP 8.5 (expressed as Present-Future RCP 8.5); (4) present with all past and future models of RCP 2.6 (expressed as Present–Past–Future RCP 2.6); and (5) present with all past and future models of RCP 8.5 (expressed as Present–Past–Future RCP 8.5). The stacked maps were generated using the software ArcGIS v. 10.5 (ESRI, Redlands, CA, USA) and modified using Canvas 12 (ACD Systems of America, Inc., Seattle, WA, USA). Map layers were obtained from site www.gadm.org.

## Data availability
The authors declare that the data supporting the findings of this study are available within the paper, its Supplementary Information and Supplementary Data files. The raw data of relict species occurrences are available from the data sources listed in Supplementary Data 4 and the corresponding authors on reasonable request; however, for the occurrences at georeferenced locations of relict species ranked as 1st–3rd protected plants in China are available from the corresponding authors

only if readers obtained permission from the Ministry of Environmental Protection of China.

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

## Acknowledgements

We acknowledge funding by the Global Environmental Research (S-14) of the Ministry of the Environment (Japan), the Proyecto Intramural Especial, PIE (grant no. 201630I024) from the CSIC, Spain, the Natural Science Foundation Project of CQ CSTC (cstc2016jcyjA0379) (China), and the Northeastern Research Institute of Petrified Wood and Mineral Resources, Nakhon Ratchasima Rajabhat University (Thailand).

## Author Contributions

C.Q.T. designed the study and wrote the manuscript. J.L.-P. wrote methodology on modeling, and contributed ideas on presenting data. M.T. and H.O. built the models and performed further analysis on the models' validations. Y.-F.D. produced the maps. M.O. contributed ideas on presenting data. S.H-M. calculated the distribution areas of the species. A.M. and B.L. interpreted the fossil data. M.W. edited the draft. C.Q.T., T.M., H.O., Y.-F.D., A.M., S.H.-M., S.Q., Y.Y., M.O., H.T.L., P.J.G., P.V.K., B.L., M.W., K.R., C.H., C.-Y.W., M.-C.P., X.C., H.-C.W., W.-H.S., R.Z., S.L., L.-Y.H., K.Y., M.-Y.Z., J.H., R.-H.Y., W.-J.L., M.T., Z.-L.W., H.-Z.Y., G.-F.Z., H.H., S.-R.Y., H.G., K.S., D.S., X.-S.L., Z.-Y.Z., P.-B.H., L.-Q. S., D.-S.H., K.L., J.L.-P. collected and analyzed the data, and jointly revised the draft.

## Additional information

**Competing interests:** The authors declare no competing interests.

Cindy Q. Tang[1], Tetsuya Matsui[2], Haruka Ohashi[2], Yi-Fei Dong[1], Arata Momohara[3], Sonia Herrando-Moraira[4], Shenhua Qian[5], Yongchuan Yang [5], Masahiko Ohsawa[6], Hong Truong Luu[7], Paul J. Grote[8], Pavel V. Krestov [9], Ben LePage[10,11], Marinus Werger[12], Kevin Robertson[13], Carsten Hobohm[14], Chong-Yun Wang[1], Ming-Chun Peng[1], Xi Chen [1], Huan-Chong Wang[15], Wen-Hua Su[1], Rui Zhou [1], Shuaifeng Li[16], Long-Yuan He[17], Kai Yan[18], Ming-Yuan Zhu[18], Jun Hu[19], Ruo-Han Yang[20], Wang-Jun Li[21], Mizuki Tomita[22], Zhao-Lu Wu[1], Hai-Zhong Yan[1], Guang-Fei Zhang[1], Hai He[23], Si-Rong Yi[24], Hede Gong[25], Kun Song[26], Ding Song[27], Xiao-Shuang Li[28], Zhi-Ying Zhang[1], Peng-Bin Han[1], Li-Qin Shen[1], Diao-Shun Huang[1], Kang Luo[29] & Jordi López-Pujol[4]

[1]Institute of Ecology and Geobotany, Yunnan University, 650091 Kunming, China. [2]Forestry and Forest Products Research Institute, Forest Research and Management Organization, Matsunosato 1, Tsukuba-shi, Ibaraki-ken 305-8687, Japan. [3]Graduate School of Horticulture, Chiba University, 648 Matsudo, Chiba 271-8510, Japan. [4]Botanic Institute of Barcelona (IBB, CSIC-ICUB), Passeig del Migdia s/n, Barcelona 08038 Catalonia, Spain. [5]Key Laboratory of Three Gorges Reservoir Region's Eco-Environment, Ministry of Education, Chongqing University, 400045 Chongqing, China. [6]The Nature Conservancy Society of Japan, Mitoyo Bldg. 2F, 1-16-10 Shinkawa, Chuo-ku, Tokyo 104-0033, Japan. [7]Southern Institute of Ecology, Vietnam Academy of Science and Technology, Ho Chi Minh City, Vietnam. [8]Northeastern Research Institute of Petrified Wood and Mineral Resources, Nakhon Ratchasima Rajabhat University, Nakhon Ratchasima 30000, Thailand. [9]Botanical Garden-Institute FEB RAS, Makovskii Str. 142, Vladivostok, Russia 690024. [10]Pacific Gas and Electric Company, 3401 Crow Canyon Road, San Ramon, CA 94583, USA. [11]The Academy of Natural Science, 1900 Benjamin Franklin Parkway, Philadelphia, PA 19103, USA. [12]Plant Ecology & Biodiversity, Utrecht University, Domplein 29, Utrecht 3512 JE, Netherlands. [13]Tall Timbers Research Station and Land Conservancy, 13093 Henry Beadel Drive, Tallahassee, FL 32312, USA. [14]Interdisciplinary Institute of environmental, Social and Human Studies, University of Flensburg, Flensburg, Germany. [15]Institute of Botany, Yunnan University, 650091 Kunming, Yunnan, China. [16]Research Institute of Resource Insects, Chinese Academy of Forestry, 650224 Kunming, China. [17]Kunming Institute of Forestry Exploration and Design, The State Forestry Administration of China, 650216 Kunming, China. [18]Centre for Mountain Ecosystem Studies, Kunming Institute of Botany-CAS, 650204 Kunming, China. [19]CAS Key Laboratory of Mountain Ecological Restoration and Bioresource Utilization & Ecological Restoration and Biodiversity Conservation Key Laboratory of Sichuan Province, Chengdu Institute of Biology, Chinese Academy of Sciences, 610041 Chengdu, China. [20]Kunming Agrometeorological Station of Yunnan Province, 650228 Kunming, China. [21]Guizhou University of Engineering Science, 551700 Bijie, China. [22]Tokyo University of Information Sciences, 4-1 Onaridai Wakaba-ku, Chiba 265-8501, Japan. [23]College of Life Sciences, Chongqing Normal University, Shapingba, 401331 Chongqing, China. [24]Chongqing Three Gorges Medical College, 404120 Chongqing, China. [25]School of Geography, Southwest China Forestry University, 650224 Kunming, China. [26]School of Ecological and Environmental Sciences, East China Normal University, 200241 Shanghai, China. [27]Kunming University of Science and Technology, 650500 Chenggong, China. [28]Yunnan Academy of Forestry, 650201 Kunming, China. [29]Key Laboratory of Tropical Forest Ecology, Xishuangbanna Tropical Botanical Garden, Chinese Academy of Sciences, Ailaoshan Station for Subtropical Forest Ecosystem Studies, National Forest Ecosystem Research Station at Ailaoshan, 650091 Kunming, Yunnan, China

