## [Peer Review File · Nature Communications]

Reviewers' Comments:

Reviewer #2:

Remarks to the Author:

General comments:

The manuscript entitled "Relict Plant Species in East Asia: Identifying Global-Stable Refugia" authored by Tang et al. addressed a very interesting and timely important topic. The authors identified the areas serving as climatic Refugia, from the past to the present and to the future, for a large number of relict plant species. In addition, areas for effective species conservation were also recommended. This study represents, to my best knowledge, the most comprehensive (in term of number of species investigated and spatial and temporal scales) study of this kind. The methodology applied is solid and the major conclusion are novel. This work constitutes a significant contribution to the scientific literature. The results were well presented and convincing, and would be interesting to a broad audience. The manuscript was well written and easy to follow. I truly enjoyed reviewing it. However, I have some minor concerns on some details (see below). Thus, I would recommend for publication in Nature Communications with minor revision.

Minor concerns:

Lines 139-140: I found the description of "the mountains in the areas stretching from near the boundary between southwestern China and northern Vietnam to central Japan" does not well represent what is shown in Figure 1. If I did not look at the map and just read this sentence, it would give me the impression of a band stretching from the west to the east instead of from the south to the north.

Lines 281-282: "while suitable areas as a whole are expected to expand generally northward for both endemic and disjunct genera". This does not seem to be true according to what is shown in Figure 5. The maps do not show a clear northward expansion, but only show contractions in the south ends. This result shown in Figure 5 are similar to what we found in projections for Chinese fir and Mason pine in China in our recent study (Wang et al. 2016). We did not find a clear northward expansion under the future climates as expected for species in North America and Europe. Our explanation for this phenomenon is the limited changes in projected precipitation under all climate changes scenarios. Thus, the distinct precipitation patterns between southern and northern China would remain unchanged, and the species currently only adaptable to the climate in southern China would probably not able to move to the north. I suggest the authors re-consider the interpretation of the results shown in Figure 5.

Line 304: The word "increase" is not a good choice here, I think. If all compared to the past, I would use the word "increase" or "decrease". In this case, I would suggest using "would be larger" instead.

Lines 334-336: "Our results highlight that on a large geographic scale, the survival and richness patterns of relict species and their forests in East Asia have been shaped by climate changes." This is a very important conclusion and I hope it appears in the abstract.

Line 660: The authors used term Ecological Niche Modeling (ENM) throughout most part of the manuscript, instead of using the term Species Distribution Modelling (SDM). I like their choice because an ecological niche or a climatic niche of a species does not represent the actual distribution of the species, which is a main reason for the climate niche based SDMs being widely criticized. However, the authors randomly used term SDMs at the end of the manuscript without defining it (Line 660). I suggest changing them back to ENM to make it consistent and to avoid confusion.

Reference

Wang, T., G. Wang, J. Innes, C. Nitschke and H. Kang (2016). "Climatic niche models and their consensus projections for future climates for four major forest tree species in the Asia–Pacific region." *Forest Ecology and Management* 360: 357-366.

Tongli Wang, PhD
Assistant Professor
Associate Director, Centre for Forest Conservation Genetics
Department of Forest and Conservation Sciences
University of British Columbia
<http://profiles.forestry.ubc.ca/person/tongli-wang/>
<http://cfcg.forestry.ubc.ca/people/tongli-wang/>
Phone: 1 604 822 1845

Reviewer #3:

Remarks to the Author:

This manuscript represents a huge amount of data compilation, fieldwork, computing, etc. It is a huge manuscript with text, and extensive methods and supplementary information. The 6 text figures contain 38 panels and the extended data have 12 figures containing 65 items. The supplementary material contains 12 tables, several of which are very large.

Because of the huge amount of information presented in various ways (text, figures, tables, supplementary material, extended data), the manuscript is challenging to read, to digest, and to understand. At times it seems to be repetitive and difficult to follow as there are many groups of taxa discussed. Only the most dedicated reader will fight /her/his way through it all. This would be a pity as it is an important contribution. Personally I think it is just too 'heavy', large, and reader-demanding to be squeezed into a Nature Communications paper.

Specific comments

l 63-65: This has been known since the days of Asa Gray, Clement Reid, et al. about 100 years ago.

l 69-70: I am not sure what the difference is between 'potential habitats' and 'suitable areas'

l 76: ice-sheets, rather than 'ice-shields'

l 80: Does 'many' refer to taxa or 'refugia areas' in the previous sentence?

l 91: elevational, rather than 'altitudinal'

l 95: Not really 'boreal' coniferous forests – they are coniferous forests but not boreal.

l 103, 109: Refugia are usually areas whose biota could grow and survive during adverse or unfavourable regional environmental conditions (e.g. arid cold glacial-stage climates) (Birks HJB 2015. *Biodiversity*. 16:196-212. doi 10.1080/14888386.2015.1117022; Birks HJB, Willis KJ 2008. *Plant Ecology and Diversity*. 1:147-160; Willis KJ, Whittaker RJ 2000. *Science*. 287:1406-1407). A climate refugium is, I believe, a new concept – see Willis and Whittaker 2000 for a discussion on the meanings of the term 'refugium'.

l 113: broad-scale, not 'large scale'

l 114, 116: Although correct English, 'emblematic' and 'depict' are unusual in a scientific text.

l 130: gymnosperms, not 'gymnosperm'

l 134: What is rarity-weighted richness?

l 143-146: I am not sure what is being correlated here. As Figure 1 is entirely maps, the data probably being correlated are spatial data and immediately questions of spatial autocorrelation arise. Has spatial autocorrelation been considered and allowed for in estimating r^2 ?

l 147, 156, 161: How is paleoendemic defined? Paleoendemic species (144) are not mentioned before.

l 193, 544: What is 'oligotypic'?

l 198: 'diminish an intensive competition' – clumsy wording

l 199: 'stress-tolerant type' – sensu Grime?

l 197-204: I feel these sentences are in the wrong order – start with habitats, then move to moderate disturbances.

l 214: Endemic today – they were once not endemic to East Asia.

l 222: The Bering land-bridge existed in the LGM.

l 238: Add a hyphen between moisture requiring so that it makes sense.

l 250: Which models? Are these for today?

l 259: They should be consistent as if I understand your modelling, these models are based on the same data, namely today's known occurrences.

l 264: Remove the comma after 'area'.

l 276-277 and elsewhere: Two lines of references to many figures and tables make reading and following the text very difficult.

l 283: 'accessible' – to what?

l 291: There is much more to an ecological niche than climate, e.g. biotic interactions, soil.

l 334: broad scale, not 'large scale'

l 336: Shaped by climate, not so much climate change.

l 343: A citation for -30°C is needed)

l 344: Better to say evolved than 'generated'

l 384: Remove the comma after 'refugia'

l 417: May, not 'Mary'

l 548: intraspecific, rather than 'infrspecific'

l 593: square, not 'squares'

l 625: What is 'background bias file'?

l 645: cross-validation, not 'cross-validate'

Extended data Figures 4 and 8: Difficult to read

Extended data Figure 12: Not easy to follow.

Tables S5-S8: It would be useful to have elevational range included

Table 59: elevational range, not 'altitudinal range'

Reviewer #4:

Remarks to the Author:

This manuscript used the occurrence of the assumed relic species to model their last glacial distribution and predict their future distributions. Based on these results and their fossil records, they concluded southwestern China and northern Vietnam served as global-stable refugia to reserve most ancient lineages. These regions have long been treated as refugia for relic genera (e.g. Zhang AL, Wu SG, eds. Floristic characteristics and diversity of East Asian plants. Beijing: Springer-Verlag, Hongkong China Higher Education Press) and the claim that Eastern Asia as refugia for relic genera is not so novel. Global-stable refugia is a novel claim, but such conclusion have some reasoning problems. Therefore, the major contributions of the present ms is to compile distribution of the species in the 'assumed' relic lineages with ancient origin and widespread previous distributions and predict the future distributions of the species from these relic lineages (see weaknesses listed in the following). However, I am mainly confused with definition of the 'refugia', 'relic species' as well as the logical reasoning of Identifying Global-Stable Refugia.

First, how to define refugia? In the current and future, distribution of most 'relic' species have been increasing with increasing temperature since the Quaternary glacial ages. Should such enlarged

distributions be called refugia? In addition, if the distributional decrease by human activity should be considered, how do you differentiate these two different 'refugia'? Why do not these relic species occupy the former widespread distributions (for example, Europe and North America) in the current and future with increasing temperature if their distributions were shaped mainly by climates? In fact, the ms seems mainly to assume their restricted distributions in Eastern Asia arose from climates (because all distributions were made based on climatic factors). But I do not believe that their distributional reductions should be only attributed to climatic cooling. If this is not the case, the reasoning to identify the current and future refugia based on niche modelling seems unreasonable. Second, as pointed out by authors in the ms, fossils could only be identified at the genus level. Therefore, genera are 'relic' (ancient origin with widespread distribution in the history but restricted distribution in the current) but species may not be true because the current species may have originated more recently (>5 millions) as for most gymnosperm species. Therefore, these species may not be the same ones as those fossils. The identified refugia based on fossils are the 'genera refugia', but the refugia identified by niche modelling are 'species refugia'. How do you scale them and unite them into a scale and concept?

Further concerns include methods and conclusions as following:

1. In the method part, I found the grid size used in the study is 1 by 1 degree. But how can these occurrence data be used in species distribution modelling with climate data at 5 by 5 km grid size? If you down-scaled the 1 by 1 degree to 5km, it definitely overestimated species occurrence.

2. The author wrote "Data were scrutinized for misidentified," (line 572), What are your rules for scrutinizing data? Except for misidentified data, specimens data also have other problems, e.g. the wrong recorded latitude and longitude or wrongly recorded localities, the precision of the occurrence data and unavailable cultivation status. The author did not described the details of the occurrence data, e.g. the time of retrieval data online, the number of total number of specimens, how many occurrences for each species and even the literature list for recording species occurrence are not been shown.

3. Line 593, "Among cells (a cell=a grid, i.e. a squares of 1° latitude × 1° longitude), we consider, the top 20%, those having species richness of at least two consecutive cells, to be the core areas of relict species richness." This is the way how the author defined core areas of relict species richness. However, the 20% threshold seems quite arbitrary. How the threshold influence the results? In fact Various criteria have been used to quantify biodiversity hotspots in previous studies, what are the differences or congruence among different criteria?

4. How did you select species specific environmental set?

Line 624: How were the candidate models built? What kind of correlation analysis and software were used?

Line 628: What do you mean candidate combinations of parameters?

Line 630: what is β -multipliers?

5. The author only used MaxEnt method for species distribution modeling, it overlooked the uncertainty SDM algorithms, especially when the species occurrences are few. At least one more SDM algorithm should be run to ensure the current results and conclusions.

6. Line 646: It is impossible to run ten replicates of SDM for when species occurrences less than 10, which probably lead to a wrong statistical results.

7. Line 654: To account for the uncertainty of SDMs, the author used the method in reference 14, which are not be statistically tested and compared with other method. So it maybe hard to say if it is better than other method or introduce more uncertainties. I prefer to use method proposed by Marmion et al. 2009, *Diversity and Distributions*, 15, 59-69.

8. Line 665 to 667: Simply intersection of potential species distribution between present and past or future is not appropriate because of the lack of real data of past or future distribution. The current species distribution could only be the migration results since LGM. Fossils of LGM might be helpful to identify the persisting areas for some species. Compared to the short lived herbs, tree species with long longevity are likely to persist in the future.

We respond to each comment of the three reviewers as follows. In the article file and Supplementary Note 1, we have highlighted the revised parts in the text.

Our response to Reviewer #2

Reviewer #2 (Remarks to the Author):

General comments:

The manuscript entitled “Relict Plant Species in East Asia: Identifying Global-Stable Refugia” authored by Tang et al. addressed a very interesting and timely important topic. The authors identified the areas serving as climatic Refugia, from the past to the present and to the future, for a large number of relict plant species. In addition, areas for effective species conservation were also recommended. This study represents, to my best knowledge, the most comprehensive (in term of number of species investigated and spatial and temporal scales) study of this kind. The methodology applied is solid and the major conclusion are novel. This work constitutes a significant contribution to the scientific literature. The results were well presented and convincing, and would be interesting to a broad audience. The manuscript was well written and easy to follow. I truly enjoyed reviewing it. However, I have some minor concerns on some details (see below). Thus, I would recommend for publication in Nature Communications with minor revision.

Minor concerns:

Lines 139-140: I found the description of “the mountains in the areas stretching from near the boundary between southwestern China and northern Vietnam to central Japan” does not well represent what is shown in Figure 1. If I did not look at the map and just read this sentence, it would give me the impression of a band stretching from the west to the east instead of from the south to the north.

The sentence is now revised as “the mountains in the areas stretching from near the boundary between southwestern China and northern Vietnam to the subtropical/warm-temperate regions of China, then to central Japan are characterized by high topographic heterogeneity and generally display an extremely high relict species richness and degree of endemism.

Lines 281-282: “while suitable areas as a whole are expected to expand generally northward for both endemic and disjunct genera”. This does not seem to be true according to what is shown in Figure 5. The maps do not show a clear northward expansion, but only show contractions in the south ends. This result shown in Figure 5 are similar to what we found in projections for Chinese fir and Mason pine in China in our recent study (Wang et al. 2016). We did not find a clear northward expansion under the future climates as expected for species in North America and Europe. Our explanation for this phenomenon is the limited changes in projected precipitation under all climate changes scenarios. Thus, the distinct precipitation patterns between southern and northern China would remain unchanged, and the species currently only adaptable to the climate in southern China would probably not able to move to the north. I suggest the authors re-consider the interpretation of the results shown in Figure 5.

Regarding the relevant content, we have revised it as “As compared to the present, in the future (2070) under six scenarios (Methods), the predicted suitable core areas with

the highest richness (51-128) of relict species will have a loss (27.8% and 68.1% on average for relict species of endemic and disjunct genera, respectively), which affects mainly their southern ends. Suitable areas with moderate richness (11-50) also show slight contractions in southern parts of the study region. Such areas of moderate richness would be generally more or less expanded northward (e.g. northeastern China, North Korea, northern Japan, and southern Kamchatka of the Russian Far East) for both endemic and disjunct genera (Figs. 4A-4N; Supplementary Tables 10 and 11), though the northward expansion would not be very distinct.”

Line 304: The word “increase” is not a good choice here, I think. If all compared to the past, I would use the word “increase” or “decrease”. In this case, I would suggest using “would be larger” instead.

We use “would be larger” instead of “increase”.

Lines 334-336: “Our results highlight that on a large geographic scale, the survival and richness patterns of relict species and their forests in East Asia have been shaped by climate changes.” This is a very important conclusion and I hope it appears in the abstract.

Yes. We have added it in the Abstract section of the revision.

Line 660: The authors used term Ecological Niche Modeling (ENM) throughout most part of the manuscript, instead of using the term Species Distribution Modelling (SDM). I like their choice because an ecological niche or a climatic niche of a species does not represent the actual distribution of the species, which is a main reason for the climate niche based SDMs being widely criticized. However, the authors randomly used term SDMs at the end of the manuscript without defining it (Line 660). I suggest changing them back to ENM to make it consistent and to avoid confusion.

Yes. We have replaced ‘SDMs’ by ‘ENMs’.

Our response to Reviewer #3

Reviewer #3 (Remarks to the Author):

General comments:

This manuscript represents a huge amount of data compilation, fieldwork, computing, etc. It is a huge manuscript with text, and extensive methods and supplementary information. The 6 text figures contain 38 panels and the extended data have 12 figures containing 65 items. The supplementary material contains 12 tables, several of which are very large.

Because of the huge amount of information presented in various ways (text, figures, tables, supplementary material, extended data), the manuscript is challenging to read, to digest, and to understand. At times it seems to be repetitive and difficult to follow as there are many groups of taxa discussed. Only the most dedicated reader will fight /her/his way through it all. This would be a pity as it is an important contribution. Personally I think it is just too ‘heavy’, large, and reader-demanding to be squeezed into a Nature Communications paper.

We have reorganized the figures and deleted previous Extended Data Fig. 8. Now there are 6 text figures and 9 supplementary figures (12 Extended Data figures as supplementary figures in our previous submission). We have deleted the section “Relict

genera: their distribution in space and time before the LGM” from the main text, but provide it as Supplementary Note 1.

Specific comments

l 63-65: This has been known since the days of Asa Gray, Clement Reid, et al. about 100 years ago.

We have revised the sentence as “We geographically delineate areas of East Asia with a sufficiently large set of data and with accurate spatial resolution to demonstrate that today Paleogene-Neogene and older relict species are abundant in...”

l 69-70: I am not sure what the difference is between ‘potential habitats’ and ‘suitable areas’

We have replaced “potential habitats” by “potentially suitable areas”.

l 76: ice-sheets, rather than ‘ice-shields’

We use “ice-sheets” instead of “ice-shields”.

l 80: Does ‘many’ refer to taxa or ‘refugia areas’ in the previous sentence?

We have revised the sentence as “Today, many of these relic lineages (some were relatively speciose such as *Gingkoales*⁶) are reduced to a few or even to a single species (Supplementary Figs. 1A-1L).”

l 91: elevational, rather than ‘altitudinal’

We have changed “altitudinal” to “elevational”.

l 95: Not really ‘boreal’ coniferous forests – they are coniferous forests but not boreal.

We changed “boreal coniferous forests” to “cold temperate coniferous forests of *Abies* and *Picea*”.

l 103, 109: Refugia are usually areas whose biota could grow and survive during adverse or unfavourable regional environmental conditions (e.g. arid cold glacial-stage climates) (Birks HJB 2015. Biodiversity. 16:196-212. doi 10.1080/14888386.2015.1117022; Birks HJB, Willis KJ 2008. Plant Ecology and Diversity. 1:147-160; Willis KJ, Whittaker RJ 2000. Science. 287:1406-1407). A climate refugium is, I believe, a new concept – see Willis and Whittaker 2000 for a discussion on the meanings of the term ‘refugium’.

We have revised the contents of these lines as follows: Ecological Niche Modeling (ENM) is widely used to predict species distributions and has been applied to identify climate refugia^{e.g. 14,15}. The term climate refugia has various definitions depending on the emphasis laid on the spatio-temporal scale of interest, the level of biotic organization involved, or the relationship with the core distribution range¹⁶. A refugium is “an area where distinct genetic lineages have persisted through a series of Tertiary or Quaternary climate fluctuations owing to special, buffering environmental characteristics”¹⁷, or “a geographical region that a species inhabits during the period of a glacial/interglacial cycle that represents the species’ maximum contraction in geographical range”¹⁸. Refugia can also be defined as “areas where local populations of a species can persist through periods of unfavorable regional climate”¹⁶, and are “habitats that components of biodiversity retreat to, persist in and can potentially expand from under changing environmental conditions”¹⁹. The emphasis on long-term persistence (or persistence through time) is clearly reflected in

the definitions of Tzedakis et al.²⁰: “a location that provides suitable habitats for the long-term persistence of populations, representing a reservoir of evolutionary history”, and Birks²¹: “the geographical area where particular plant and animal populations grow today, grew in the past or may persist in the future”. ENM is an effective and meaningful way to identify refugia, as climatic-based paleodistribution reconstructions often show good agreement with other proxies such as genetic markers^{e.g. 22,23} ...

l 113: broad-scale, not ‘large scale’

We use broad-scale instead of large scale.

l 114, 116: Although correct English, ‘emblematic’ and ‘depict’ are unusual in a scientific text.

We changed “emblematic” to “symbolic” and the sentence with “depict” is deleted now in the revised MS.

l 130: gymnosperms, not ‘gymnosperm’

We use “gymnosperms” instead of “gymnosperm”.

l 134: What is rarity-weighted richness?

We have revised the sentence as “The patterns obtained for species richness and rarity-weighted richness (the latter representing concentrations of limited-range species, as well as a high turnover of species between adjacent cells²⁴) show that the relict species are mainly confined to areas to the south of the summer monsoon limit line (Figs. 1A-1D).”

l 143-146: I am not sure what is being correlated here. As Figure 1 is entirely maps, the data probably being correlated are spatial data and immediately questions of spatial autocorrelation arise. Has spatial autocorrelation been considered and allowed for in estimating r^2 ?

We have added sentences as “The correlation of species richness and rarity-weighted richness was analyzed using Pearson’s correlation coefficient. The related p -value was corrected for the spatial autocorrelation with Dutilleul’s⁵⁸ method and implemented with R Package ‘SpatialPack’ Version 0.3⁵⁹ (<http://spatialpack.mat.ufsm.cl>)” in the Method section.

In Lines 164-168 on Page 8 of the revised MS, we state that “In the vast geographic region of East Asia, the distribution patterns of relict species richness (Figs. 1A and 1C) and rarity-weighted richness (Figs. 1B and 1D) have linear relationships with significant positive correlations (Pearson’s $r = 0.83$, spatial autocorrelation-corrected $p < 0.0001$ for relict species of endemic genera; $r = 0.63$, spatial autocorrelation-corrected $p < 0.0001$ for relict species of disjunct genera).”

l 147, 156, 161: How is paleoendemic defined? Paleoendemic species (144) are not mentioned before.

We have revised the sentence as “The most important large-size core area with the highest paleoendemic species (species that were formerly widespread but are now restricted to a smaller area) richness (144) is situated in...”

l 193, 544: What is ‘oligotypic’?

We have revised the sentence as “Some stands dominated by oligotypic genera (genera of plants that contain only a few species) having...”

l 198: ‘diminish an intensive competition’ – clumsy wording

We have changed “diminish an intensive competition” to “with little competition”.

l 199: ‘stress-tolerant type’ – sensu Grime?

Yes. We have revised it as “the stress-tolerant type^{sensu 26}”

l 197-204: I feel these sentences are in the wrong order – start with habitats, then move to moderate disturbances.

We have revised these sentences as “Among the 423 relict forests, 92% of the stands are found in mountain stream flood plains and ravines, and on scree or steep slopes, cliffs and rocky terrains or limestone areas where natural disturbances often occur. Thus, as a general rule, relict species are restricted to local habitats with little competition with non-relict species in the regeneration phase, and most of them appear to be shade-intolerant and to belong to the stress-tolerant type^{sensu 26}.”

l 214: Endemic today – they were once not endemic to East Asia.

Yes, we have added “today” after the word “endemic”.

l 222: The Bering land-bridge existed in the LGM.

Temperatures at the latitudes of the Bering Land Bridge during the Paleogene were substantially warmer than during the LGM. As such, the warm temperate elements that were able to adapt/survive under the unique polar lighting conditions were able to use the BLB. Although the BLB was functional during the LGM, temperature precluded most vegetation from the region.

We have revised the sentence as “The present endemism was formed when some dispersal routes became discontinuous, such as the North Atlantic land bridge and areas along the Tethys Seaway in the Paleogene and the much colder and drier conditions precluded the use of the Bering land bridge by temperate floral elements by late Neogene time and for all plants due to glaciation in the Quaternary^{83,84}.”

Note: In the revised MS, the section “Relict genera: their distribution in space and time before the LGM” is moved to Supplementary Note 1 from the main text.

l 238: Add a hyphen between moisture requiring so that it makes sense.

Yes. We have added a hyphen.

l 250: Which models? Are these for today?

We have revised the sentence as “All ecological niche models (Methods) performed well with both jackknife...”

l 259: They should be consistent as if I understand your modelling, these models are based on the same data, namely today’s known occurrences.

Regarding ENM, modeling is not necessarily consistent with observed species richness patterns. For example, the areas potentially suitable such as the Himalayas and boundary between NW India/Myanmar have low richness values. That means a place may be potentially suitable for a given species, but the species apparently could not reach that place (with enough propagules) to establish natural populations. We call this place as “an empty habitat”.

l 264: Remove the comma after ‘area’.

Yes. We have removed the comma.

l 276-277 and elsewhere: Two lines of references to many figures and tables make reading and following the text very difficult.

We have reorganized the figures and the relevant contents.

l 283: ‘accessible’ – to what?

We have revised it as “accessible to the relict plants”.

l 291: There is much more to an ecological niche than climate, e.g. biotic interactions, soil.

We have revised it as “...to reveal areas that remain stable in climatic variables.”

l 334: broad scale, not ‘large scale’

We have changed “large” to “broad”.

l 336: Shaped by climate, not so much climate change.

We have deleted “change”.

l 343: A citation for -30°C is needed)

We refer the data in our Fig. 6B.

l 344: Better to say evolved than ‘generated’

We used “formed” instead of “generated”. We think we’d better not to use “evolved” in order to avoid the concept ‘evolution’ in relict lineages of East Asia that, for most cases, are characterized by a clear morphological status (e.g. *Ginkgo biloba*). The word “evolved” may cause confusion and misunderstandings here.

l 384: Remove the comma after ‘refugia’

Yes. We have removed the comma.

l 417: May, not ‘Mary’

Yes. We have done the correction.

l 548: intraspecific, rather than ‘infraspecific’

In the context, we refer the taxonomic term “intraspecific”—of or relating to a subdivision of a species, as a subspecies, variety. We keep the word “intraspecific” and we delete “level”.

l 593: *square, not 'squares'*

We have changed “squares” to “square”.

l 625: *What is 'background bias file'?*

The “background bias file” is a file that allows the user to choose background data with the same bias as occurrence data. This file is used to cope with the problems associated with uneven sampling efforts (see Phillips et al., 2009, *Ecological Applications*, 19: 181-197). We add the citation after the term “background bias file” in the text.

l 645: *cross-validation, not 'cross-validate'*

We have changed “cross-validate” to “cross-validation”.

Extended data Figures 4 and 8: Difficult to read

We have revised and reorganized previous Extended Data Figures 4, 5, 6, 7. We delete previous Extended Data Figure 8.

Extended data Figure 12: Not easy to follow.

We have revised this figure, now it is Supplementary Figure 9 in the revised MS.

Tables S5-S8: It would be useful to have elevational range included

Yes. We have provided the elevational ranges in Supplementary Tables 5-8.

Table 59: elevational range, not 'altitudinal range'

We have changed “altitudinal” to “elevational”.

Our response to Reviewer #4

Reviewer #4 (Remarks to the Author):

*This manuscript used the occurrence of the assumed relic species to model their last glacial distribution and predict their future distributions. Based on these results and their fossil records, they concluded southwestern China and northern Vietnam served as global-stable refugia to reserve most ancient lineages. These regions have long been treated as refugia for relic genera (e.g. Zhang AL, Wu SG, eds. *Floristic characteristics and diversity of East Asian plants*. Beijing: Springer-Verlag, Hongkong China Higher Education Press) and the claim that Eastern Asia as refugia for relic genera is not so novel. Global-stable refugia is a novel claim, but such conclusion have some reasoning problems.*

Indeed, the claim that East Asia harbored refugia for relict plants is not new. However, this is the first study that aims to geographically delineate it with a sufficiently large set of data and with accurate spatial resolution. Also, our effort to detect southwestern China and northern Vietnam as long-term, stable refugia is totally new to our knowledge, as it allows for detecting refugia with the same species persisting across time (described in lines 662-678 in previous MS, now it is in lines 665-681 in the revised MS, also previous Extended Data Figure 12, now it is Supplementary Figure 9 in the revised MS).

Therefore, the major contributions of the present ms is to compile distribution of the species in the 'assumed' relic lineages with ancient origin and widespread previous distributions and

predict the future distributions of the species from these relic lineages (see weaknesses listed in the following). However, I am mainly confused with definition of the 'refugia', 'relic species' as well as the logical reasoning of Identifying Global-Stable Refugia.

First, how to define refugia? In the current and future, distribution of most 'relic' species have been increasing with increasing temperature since the Quaternary glacial ages. Should such enlarged distributions be called refugia?

The term ‘refugia’ has been subject to many definitions over recent years, and it is still subjected to much discussion (see e.g. Hampe, 2013, *New Phytol.* 197: 16–18; Gavin et al. 2014, *New Phytol.* 204: 37–54). A simple definition, for example, is provided in Hampe et al. (2013) as those “areas where local populations of a species can persist through periods of unfavorable regional climate”. However, as the same authors recognize, several concepts of climate refugia have arisen, many depending on a spatio-temporal scale of interest, and so can be viewed, as ‘past refugia’ (that is, glacial) as ‘present-day refugia’, and ‘putative refugia’ (the two latter being interglacial). The current and future refugia (those mentioned by the Reviewer), together with the past (LGM) refugia, are identified by our analyses (previous Figs. 4, 5, now Figs. 3 & 4). Other definitions of refugia rely on areas that harbor plant populations through time, e.g. that of Keppel et al. (2012, *Glob. Ecol. Biogeogr.* 21: 393–404): “habitats that components of biodiversity retreat to, persist in and can potentially expand from under changing environmental conditions”, or that of Tzedakis et al. (2013, *Trends Ecol. Evol.* 28: 696–704): “locations that provide habitats for the long-term persistence of populations”. This wider definition is also covered by our approximation to detect long-term, stable refugia (previous Fig. 6, and now Fig. 5).

We have added relevant contents in lines 109-122 in the revised MS.

In addition, if the distributional decrease by human activity should be considered, how do you differentiate these two different 'refugia'?

We are not able to estimate ‘future’ refugia taking into account human activity, given the fact that human activity cannot be extrapolated to the year 2070 (there is no available layer predicting how the habitats or the human footprint will be in the future). Moreover, to ensure that the past, present, and future models are comparable, and more importantly, to delimit those areas acting as long-term refugia, the environmental layers employed should be the same for each time slice.

Why do not these relic species occupy the former widespread distributions (for example, Europe and North America) in the current and future with increasing temperature if their distributions were shaped mainly by climates? In fact, the ms seems mainly to assume their restricted distributions in Eastern Asia arose from climates (because all distributions were made based on climatic factors). But I do not believe that their distributional reductions should be only attributed to climatic cooling. If this is not the case, the reasoning to identify the current and future refugia based on niche modelling seems unreasonable.

The regional extinction of these relict lineages in Europe and North America is still not well understood, but scholars often cite climate cooling as the main cause. In previous trials, areas of both Europe and North America appear as suitable for some species. However, as these extinctions took place so long ago (mainly during the Eocene, Oligocene, Miocene and Pliocene, please see previous Figs. 3 G-J, now Supplementary Fig. 6), we have not considered these two regions in our models given that the arising of new populations of

these species/lineages there would be more an entelechy than a real possibility (unless man-made reintroductions were undertaken).

Many aspects of distribution (restriction, elimination, expansion) can be impacted by biotic and abiotic factors. Examples include orogenies, epeiric seaways, plate tectonics impacting air flow patterns and ocean circulation patterns, which ultimately determine local, regional, and global climate. Ecological level impacts occur at many levels, shaping plant communities. These factors taken together drive plant evolutionary processes and determine community composition and structure, and the paleobotanical literature shows that climate (together with some other factors) has largely been responsible for plant evolution, floristic composition and distribution for the last 400 million years. To the question posed by the Reviewer – relict species occupation of former widespread distributions. Yes, they could conceivably re-occupy formerly widespread distributions, but it is unlikely, because the plants are adapted to the current biotic and abiotic conditions and would need to evolve quickly (a few millions of years) to take advantage of the new climate.

Second, as pointed out by authors in the ms, fossils could only be identified at the genus level. Therefore, genera are 'relic' (ancient origin with widespread distribution in the history but restricted distribution in the current) but species may not be true because the current species may have originated more recently (>5 millions) as for most gymnosperm species. Therefore, these species may not be the same ones as those fossils. The identified refugia based on fossils are the 'genera refugia', but the refugia identified by niche modelling are 'species refugia'. How do you scale them and unite them into a scale and concept?

Having absolute confidence as to whether all the selected species are true relicts is nearly impossible, because, as the Reviewer mentioned, it is hard to assign fossils below the genus level, and dated phylogenies are still scarce (only available for a small proportion of the species included in our study). However, as we state in lines 539-547 in the previous MS and now in lines 532-543 in the revised MS, we have used all the data available for each species, in addition to fossil and molecular phylogenies, to classify them as 'relict' species. These data included morphology (i.e. presence of primitive characteristics) as well as systematic and biogeographical data: we have excluded, for example, large genera with many species even when the genus was of relict origin (e.g. *Pinus*); we also excluded closely related taxa that are geographically clustered, as this pattern is typical of neoendemics.

The species concept has been the subject of discussion and debate since the concept was proposed and continues today. The Reviewer makes an important point that the extant species may not be the same as the fossils. This is absolutely correct, but it depends entirely on the criteria one uses to define a species. Today, we may define a species on the basis of genetics, physiology, biochemistry, etc. Paleobotanists are striving to define species using a biological species concept, but in the end, we are dealing with plants that have been dead and one can never truly know. The concept of the nearest living relative and coexistence approaches provide processes for defining species using the weight of evidence by comparing similarities/differences between fossil and living taxa. From a biogeographic perspective, consideration of the entirety of the fossil record of a genus provides insight into the species of that group that were present through time. The species can be tracked in space and time to determine responses to biotic and abiotic changes and using the characters of the living species, reasonably good assumptions and statements can be made

about the fossil taxa. It is assumed that the genus level climate requirements have changed little since the Eocene as demonstrated by Mosbrugger and Utescher (1997, *Palaeogeogr. Palaeoclimatol. Palaeoecol.* 134: 61-86) and results of climate reconstruction based on modern relatives of fossil species are very much consistent with those of abiotic proxies as oxygen isotope curves (Mosbrugger et al., 2005, *PNAS* 102: 14964-14969). The method for inferring paleoclimates using fossils' nearest living relatives-extant species has been validated by Harris et al. (2017, *Review of Palaeobotany and Palynology* 244: 316–324).

Further concerns include methods and conclusions as following:

1. *In the method part, I found the grid size used in the study is 1 by 1 degree. But how can these occurrence data be used in species distribution modelling with climate data at 5 by 5 km grid size? If you down-scaled the 1 by 1 degree to 5km, it definitely overestimated species occurrence.*

The 1° latitude by 1° longitude is the scale to show the present species richness and rarity-weighted richness maps, not the ENM (which was done at a resolution of 2.5 arc-min). The occurrences for all the species (for which ENM were built) were precise enough to allow for work at a resolution of 2.5 arc-min (occurrences with not enough geographical precision were discarded for the ENM analyses). It is stated in lines 580-581 and 621-622 of the previous MS. We further clarify this in the Methods section in the revised MS (see lines 589-590 and 622-623.

2. *The author wrote “Data were scrutinized for misidentified,” (line 572), What are your rules for scrutinizing data? Except for misidentified data, specimens data also have other problems, e.g. the wrong recorded latitude and longitude or wrongly recorded localities, the precision of the occurrence data and unavailable cultivation status. The author did not described the details of the occurrence data, e.g. the time of retrieval data online, the number of total number of specimens, how many occurrences for each species and even the literature list for recording species occurrence are not been shown.*

We have included additional details on how the localities were collected and selected in the Methods section of the revised MS. We provide Supplementary Table 15 for all the physical herbaria, websites with the time of retrieval data online, and literature that we used for our occurrence data.

There is no meaning to provide the total number of specimens, since our occurrence data are not just from specimens, they are also from our fieldwork over years and publications.

In lines 573-574 in the revised MS, we state that “For the number of occurrences for each species please see Supplementary Tables 3 and 4”.

3. *Line 593, “Among cells (a cell=a grid, i.e. a squares of 1° latitude × 1° longitude), we consider, the top 20%, those having species richness of at least two consecutive cells, to be the core areas of relict species richness.” This is the way how the author defined core areas of relict species richness. However, the 20% threshold seems quite arbitrary. How the threshold influence the results? In fact Various criteria have been used to quantify biodiversity hotspots in previous studies, what are the differences or congruence among different criteria?*

We made several trials using the top 5%, 10%, 15% and 20%, those having species richness to observe the general patterns of the core areas of relict species richness of genera endemic to East Asia. We found that the top 20%, those having species richness of at least two consecutive cells, contain all the most important species of monotypic genera including *Ginkgo*, *Metasequoia*, *Taiwania*, *Cathaya*, *Pseudolarix*, *Cryptomeria*, *Sciadopitys*, *Thujopsis*, *Cercidiphyllum*, *Davidia*, *Tetracentron* and so on. Thus we chose the top 20%, those having species richness of at least two consecutive cells, as the threshold for the core areas of relict species of genera endemic to East Asia. In the geologic past, plant community reconstructions show that many of these taxa were present a well-defined niche within the regional landscape. Trials below 20% eliminated too many taxa to be representative of a regional flora.

To keep things comparable, we use the same threshold for relict species of genera having disjunct distributions.

4. *How did you select species specific environmental set?*

From the set of the 19 bioclimatic variables provided in WorldClim (<http://www.worldclim.org/bioclim>), we selected the best subset of them for each species, using the methodology described in lines 623-633 in our previous MS, now in lines 622-633 in the revised MS.

Line 624: How were the candidate models built?

First we created all possible combinations of 19 Bioclim variables, then we excluded those models that included combinations of highly correlated variables (Pearson's $r \geq |0.70|$). Then, we calculated Variance Inflation Factors (VIF); datasets with $VIF \geq 5$ were excluded to avoid multi-collinearity. VIF were estimated using the `vif` function included in the `usdm` package in R. By this procedure, the number of final candidate combinations of parameters was 2811. We stated it in lines 625-629 of the previous MS. Now it is in lines 626-631.

What kind of correlation analysis and software were used?

We used Pearson's r for correlation analysis in R (package `base`), so the text was modified as follows:

“First, we excluded those models that included combinations of highly correlated variables (Pearson's $r \geq |0.70|$).”

Line 628: What do you mean candidate combinations of parameters?

By ‘candidate’ we mean all the possible combinations of variables that were subsequently analyzed, to look for the ‘best’ combinations (that is, those finally selected to run the niche models). To us, this terminology is broadly used in plant science and biological sciences in general (e.g., ‘candidate’ genes, or ‘candidate’ molecules).

Line 630: what is β -multipliers?

β -multipliers is equal to the regularization multiplier. It is a parameter that adds new constraints, in other words is a penalty imposed to the model. The main goal is to prevent over-complexity and/or overfitting by controlling the intensity of the chosen

feature classes used to build the model (Morales et al 2017, PeerJ, doi: 10.7717/peerj.3093). We have added the reference after “ β -multipliers” in the revised MS.

5. *The author only used MaxEnt method for species distribution modeling, it overlooked the uncertainty SDM algorithms, especially when the species occurrences are few. At least one more SDM algorithm should be run to ensure the current results and conclusions.*

We used MaxEnt because it is the most robust SDM algorithm for any sample size and at any scale (Hernandez et al., 2006, *Ecography*, 29: 773-785; Wisz et al. 2008, *Divers. Distrib.* 14: 763-773; Tognelli et al., 2009, *Rev. Chil. Hist. Nat.*, 82: 347-360; Tarkesh & Jetschke, 2012, *Environ Ecological Stat* 19: 437-457 Aguirre-Gutiérrez et al., 2013, *PLoS ONE* 8: e63708; Qiao et al., 2015, *Sci. Rep.*, 5: 14232), especially when absence data are hard to collect (which is our case, and the large majority of the cases in the SDM literature). We believe that employing a second SDM algorithm is not necessary for the present work, for the following reasons: (1) it is not expected to produce significant changes in the patterns of suitable habitats at such a large scale (e.g. Carvalho et al., 2015, *PLoS ONE* 10: e0143282); (2) the uncertainty of our models has been properly addressed by using the method described in Tang et al. 2017 (*Sci. Rep.* 7: 43822); (3) the usage of a second SDM will mean the unnecessary lengthening of the manuscript, as the number of figures and tables will be automatically duplicated. Please note that the Reviewer #3 alerts in the contrary sense (i.e., that the MS has a huge amount of analyses, tables, and figures). In fact, studies including a large amount of data such as ours and studying large regions usually use only a single SDM, and in almost all cases are choosing MaxEnt. Examples include articles recently published in *Nature Communications* in the last 5 years, e.g. Kearns et al. 2018, *Nat. Comm.* 9:906; Runting et al., 2015, *Nat. Comm.* 6:6819; Mokany et al. 2014, *Nat. Comm.* 5:3971; Brown et al., 2014, 5: 5046; Kearns et al., 2014, 5: 3994; or Ikeda et al., 3: 648.

A last (but not least important) reason is technical: running an additional SDM, given the huge number of species and the large area studied, might take many months, if not 1 year (we spent over 1 year from the running of the first models until we got the final maps shown in Figs. 3-5 and made the calculations shown in Supplementary Tables 10-13).

Note: in the revised MS we now use the terminology ENM instead of SDM.

6. *Line 646: It is impossible to run ten replicates of SDM for when species occurrences less than 10, which probably lead to a wrong statistical results.*

For the cross-validation approach, we ran ten replicates to obtain more robust modelling results. For the jackknife approach, we ran replicates with the same number of occurrence points. We have corrected this mistake regarding jackknife in the revised version of the MS. Please see lines 648-650 in the revised MS.

7. *Line 654: To account for the uncertainty of SDMs, the author used the method in reference 14, which are not be statistically tested and compared with other method. So it maybe hard to say if it is better than other method or introduce more uncertainties. I prefer to use method proposed by Marmion et al. 2009, *Diversity and Distributions*, 15, 59-69.*

We believe that our method is good for treating uncertainties, as we only consider that a given area is climatically suitable when at least 95% of the models run forecast that area as suitable. Thus, it should be regarded as a very strict method to treat model

uncertainty and, indeed, the suitable area recovered by applying our uncertainty method was reduced by ca. 37% compared with a ‘standard’ model in the test we did with *Davidia involucrata* (Tang et al. 2017, Sci. Rep. 7: 43822). Our method was published just one year ago, so we are not aware of any study comparing our method to treat uncertainty with other methods. Every author may have a preferred method and we do not say that our method is better than others. We can only say that our method ensures that only those areas with high probability of species occurrence are robustly identified, a pre-requisite in studies with strong implications for biodiversity conservation.

Please note that treating uncertainty, though highly recommendable, is not always addressed; hence, of the examples of articles included in our response to the Reviewer’s question no. 5, none of them directly addressed the model uncertainty.

8. Line 665 to 667: *Simply intersection of potential species distribution between present and past or future is not appropriate because of the lack of real data of past or future distribution. The current species distribution could only be the migration results since LGM. Fossils of LGM might be helpful to identify the persisting areas for some species. Compared to the short lived herbs, tree species with long longevity are likely to persist in the future.*

Of course, the act of projecting the current potential distribution areas to the past or to the future is based on climatic models and not real data. However, how much research is based on models and not real data (because these cannot be collected)? Are the projections of the IPCC (Intergovernmental Panel on Climate Change) invalid (or not worth doing) because they are based on projections? Or, were the studies of eminent scientists such as Albert Einstein not valid because they were mostly based on conjectures? Niche modelling is a well-established method, with thousands of articles based on this methodology published during the last decade (many of them in the best scientific journals including Nature, Science, Nature Communications or PNAS). Moreover, projections to the past with niche models usually show a very good agreement with genetic data when available, often with significant correlations when this has been statistically tested (e.g. Chung et al., 2018, PLOS One, 13(1): e0190520; Noguerales et al., 2018, Molec. Phylogenet. Evol., 118: 343–356).

Regarding the possibility that current species distribution could represent the migration of species since LGM, the effect of migration is avoided particularly by means of our approximation to detect long-term, stable refugia (as is carefully described in lines 662-677 of previous MS and now in lines 662-678 of the revised MS, see also previous Extended Data Fig. 12, now supplementary Fig. 9 in the revised MS).

As stated below, and recognized by the Reviewer, it is hard to assign fossils below the genus level.

Yes, in general tree species have greater longevity than herbs; however, if the climate becomes unsuitable in a given region, trees, as herbs, will quickly decrease their fitness and may die after a short period of time. Many instances are documented of death en mass of trees because of drought or increased temperatures; for example, a paper published in PNAS (102: 15144-15148) in 2005 reports that >90% of the dominant, overstory tree species (*Pinus edulis*, a piñon) in southwestern North American woodlands died only after 15 months (in 2002-2003) of global-change-type drought. Please see Allen et al. 2010 (Forest Ecol. Manag. 259: 660–684) for a global assessment of recent tree mortality

attributed to drought and heat stress, which, according to the authors, mean that the world's forested ecosystems already may be responding to climate change.

Reviewers' Comments:

Reviewer #2:

Remarks to the Author:

My comments and suggestions have been well addressed in the revised manuscript. I have no further concern.

Reviewer #4:

Remarks to the Author:

Although authors listed some publications, they did not see the potential caveats in these published papers. It is better to point out caveats of the modelling the distributions. I appreciate the data collected here and the modelling results obtained. However, the main scientific logical reasoning and conclusions are still difficult to follow. Therefore, more revisions, especially on using the terms 'refugia' in the expressions of the future distributions, are still needed. Although authors listed diverse definitions of refugia, they did not point out the common essential of these definitions: refugia is defined based on the contrasted distributions of one species (or one flora) at different stages because of the climatic oscillations. SW China is considered as the refugia of these ancient lineages at LGM or the current, because before LGM and current, they had distinctly larger distributions. At the current or aft LGM with warmer climate, if distributions of these relic species are determined by climate, why did not they expand as numerous species? The limited distributions of these relic species may be mainly shaped by their limited evolutionary potentials (greatly reduced in dispersal and adaptation because of the too small population sizes of these endangered relic species) and human activities. Therefore, the term 'refugia' coined for climatic oscillations of different stages and modelling researches are really difficult to feel comfortable under such a scenario.

1. One similar research on plants in southwest China also used the similar niche modelling way (in fact more complex and better) to predict the distribution changes of plants in this region in response to the climate changes (Liang et al. 2018, Journal of Biogeography, DOI: 10.1111/jbi.13229). They pointed out many caveats in such modelling results.
2. In the future, these ancient species still persist in south west China although shifting southward. As pointed out by authors 'the areas as a whole will probably expand'. Therefore, based on these range changes as well as the climatic warming, it should be very careful to conclude about the 'permanent' refugia. Under this scenario, as responded by authors, even if refugia may have different definitions (however, always against the clear backgrounds, for example, distributions in LGM versus pre- or post-LGM), it is still difficult to reason that future distributions are 'refugia'. What is the background for refugia in the future? Are you assuming: these ancient should be distributed in the whole northern Hemisphere in the current or in 2070? Therefore I could not follow their logic conclusions.
3. In the Liang et al. (2018) paper, they compared the range sizes of all sampled species in the current, LGM and 2050. I am wondering whether the claim 'the areas as a whole will probably expand' should include meta-analyses of range size changes of the sampled ancient species during different stages. This will give a clear over-view of the distributions of these ancient lineages from LGM, the Current and 2070. If the overall distributions are expanding as similar to those species examined by Liang et al. (2018), the conclusion 'stable reguia' is not good to express this pattern. If the overall distributions are shrinking from LGM to the Current to 2070, you can say that 'this region' is a 'refugia' against the current or LGM distributions. Overall, I feel so weird to say 'refugia' when without distinct evidence for contrasted distribution range sizes between different stages.
4. I feel that it is better not to say 'the survival of these relic species was shaped by climate in Asia'. It is very obvious that most of these relic species could grow very well in European gardens, for example, Davidiana and other species. In fact, well growth and normal reproductions of these relic

species in Europe and North America raised further niche modelling questions on their conclusions and assumptions. At least, the limitations for such modelling and those existing facts should be mentioned.

5. Another question related to the above questions, is that most of these relic or endangered species are also widely cultivated in Asia because of their great values in conservation, re-forestation, researches and others. I did not find the details that how they differentiated the natural distributions and artificial cultivation. If you google, you will find so many sites where such one relic species (for example, Davidiana, Ginkgo, Metasequoia) was cultivated.

6. Another finding that rich relic flora occurred in the eastern Asia (isoline by precipitation) is similar to that recently reported in Nature (Lu et al. 2018, evolutionary history of the angiosperm flora of China). IN this nature paper, they also pointed out many relic lineages occurred in the eastern Asia. This work should be cited. They also mentioned that SW China served as the refugia for numerous ancient lineages. However, their conclusions were based on the current limited distributions with the ancient wide distributions. This logic expression can be followed.

Dear Dr. McKay,

We've revised the manuscript following your suggestions, specifically to the two points you brought up and have incorporated them in this 2nd revision of our manuscript. We briefly reply to Reviewer #4's comments and provide our responses in this letter.

Our responses to the Editor:

Your manuscript entitled "Relict Plant Species in East Asia: Identifying Long-Term Stable Refugia" has now been seen by Reviewers 2 and 4 (Reviewer 3 was unavailable). You will see from their comments below they continue to raise some important points which must be addressed in the form of a revised manuscript before we make a final decision on publication. We therefore invite you to revise and resubmit your manuscript, taking into account the points raised. In particular, we request that two remaining issues be addressed through text revision:

(1) Please further clarify the definition of refugia being used here compared to the literature, and please ensure that the definition remains internally consistent throughout the paper (e.g. in both the hindcasting and forecasting aspects).

The definition of “refugia” is broad and has been used in many different ways through time by the scientific community. Even today, the large number of papers (e.g., Bennett & Provan, 2008, *Quat. Sci. Rev.*, 27: 2449–2455; Holderegger & Thiel-Egenter, 2009, *J. Biogeogr.*, 36: 476–480; Rull 2009, *J. Biogeogr.*, 36: 481–484; Ashcroft, 2010, 37: 1407–1413; Stewart et al., 2010, *Proc. Royal Soc. B, Biol. Sci.* 277, 661–671; Keppel et al., 2012, *Glob. Ecol. Biogeogr.* 21, 393–404; Hampe et al., 2013, *New Phytol.* 197, 16–18; Tzedakis et al., 2013, *Trends Ecol. Evol.* 28, 696–704; Gavin et al., 2014, *New Phytol.*, 204, 37–54; Birks, 2015, *Biodiversity*, 16, 196–212) published in reputable international journals that continue to focus on the definition of refugia indicates this is a continuing matter of debate and perhaps a little contentious.

The definition used by us in this paper (which is based on the definitions of Tzedakis et al., 2013, *Trends Ecol. Evol.* 28, 696–704, and Birks, 2015, *Biodiversity* 16, 196–212) was clearly stated in the Introduction in our 1st revision and in this reversion we have clarified it further by adding the following sentence after the sentences that indicate the term refugia has been modified and used in many different contexts by the scientific

community: *In the present study we adopt the prerequisite of long-term persistence as the main trait defining refugia and, for the context of relict plant species in East Asia, we define “long-term stable refugia” as the climatically suitable areas that allowed the persistence (in contrast to other areas) of ancient lineages during the Pleistocene climatic oscillations and that probably will do under a scenario of global warming. We also introduced a clarification between brackets in the last sentence of the Introduction, as follows: Finally, we identify long-term (at least since the LGM to the year 2070) stable refugia as maintaining ancient lineages and propose establishment of protected areas for long-term stable refugia.*

We also made sure that use of “long-term stable refugia” in the ms. (and the more general term “climate refugia”) remains internally consistent throughout the paper.

In the responses to Reviewer #4 we have also clearly stated that the term “refugia” is only applied to the results shown in Fig. 5, while those shown in Figs. 3 and 4 only illustrate of the potential richness of the studied relict species in the past (LGM and Holocene), present, and future (year 2070). Please note that the word “refugia” is not mentioned in the entire section of the paper “Relict species richness patterns shaped by climate changes”. We can only conclude that Reviewer #4 did not understand these concepts.

(2) Please further acknowledge and describe and the assumptions and caveats underlying conclusions about species past and future ranges. For example, it should be clear that ENMs are only estimates of the realized niche, which could be influenced by numerous factors other than climate.

We understand that our ms. contains some assumptions and caveats underlying the conclusions, although the existence of such assumptions and caveats is a common issue in all modelling studies. Nature is very complicated and the scientific community as a whole continues to struggle to understand the complex relationships and interactions between elements of each ecosystem. Clearly, there is no single model that can model/predict the entirety of a functional ecosystem. We do the best that we can with working with various elements that we understand and hope these contributions will improve understanding of these systems. We are well aware of the

limitations and caveats of our results and following the Editor's suggestion, acknowledged them as much as we could, including the example provided. In addition to small changes along the text, in the Introduction section we have added an entire paragraph regarding the caveats and assumptions in ENM studies:

“It should be noted, nevertheless, that conclusions based on ENM rely on a series of basic assumptions (e.g. Araújo & Peterson, 2012) that, if not met, may compromise in a certain degree the validity of the results found: (1) niches are conserved along time (i.e. “niche conservatism”; Wiens & Graham, 2005), and (2) a given species has access to all possible environmental conditions, unrestricted by barriers, dispersal disequilibrium, or negative interactions; that is, that fundamental niche can be equated to the realized niche (Soberón & Peterson, 2005). While niche conservatism may apply in many cases (Peterson, 2011), realized niches are generally a subset of fundamental ones (Soberón & Arroyo-Peña, 2017). Since ENM is calibrated with the observed distribution of the species, a given modelled niche actually corresponds to the realized niche and, thus, the fundamental niche could be shaped by other factors than environment (e.g. biotic interactions with other species, dispersion ability, and other abiotic factors). Despite these limitations, ENM is still a very powerful approach that is not only widely used to detect refugia, but also to discover new populations or species, to determine the impact of invasive species, to predict areas with conservation ends (for designing protected areas, or areas for restoration, translocation, or reintroductions), to evaluate the impacts of climate change on biodiversity, or to ask questions regarding the patterns of niche evolution (Araújo & Peterson, 2012).”

But we would like to point out that, if we are attaching more importance than deserved to these assumptions and caveats (as it seems that Reviewer #4 is particularly concerned), this will make research impossible/unworkable.

Our responses to Reviewer #2:

My comments and suggestions have been well addressed in the revised manuscript. I have no further concern.

Thank you very much for your expertise!

Our responses to Reviewer #4:

Although authors listed some publications, they did not see the potential caveats in these published papers. It is better to point out caveats of the modelling the distributions. I appreciate the data collected here and the modelling results obtained. However, the main scientific logical reasoning and conclusions are still difficult to follow.

Following Reviewer #4's suggestions, we have pointed out potential caveats concerning distribution models (as much as we could). In addition to small changes in the text, in the Introduction section we have added an entire paragraph summarizing the caveats and assumptions in ENM studies:

“It should be noted, nevertheless, that conclusions based on ENM rely on a series of basic assumptions (e.g. Araújo & Peterson, 2012) that, if not met, may compromise in a certain degree the validity of the results found: (1) niches are conserved along time (i.e. “niche conservatism”; Wiens & Graham, 2005), and (2) a given species has access to all possible environmental conditions, unrestricted by barriers, dispersal disequilibrium, or negative interactions; that is, that fundamental niche can be equated to the realized niche (Soberón & Peterson, 2005). While niche conservatism may apply in many cases (Peterson, 2011), realized niches are generally a subset of fundamental ones (Soberón & Arroyo-Peña, 2017). Since ENM is calibrated with the observed distribution of the species, a given modelled niche actually corresponds to the realized niche and, thus, the fundamental niche could be shaped by other factors than environment (e.g. biotic interactions with other species, dispersion ability, and other abiotic factors). Despite these limitations, ENM is still a very powerful approach that is not only widely used to detect refugia, but also to discover new populations or species, to determine the impact of invasive species, to predict areas with conservation ends (for designing protected areas, or areas for restoration, translocation, or reintroductions), to evaluate the impacts of climate change on biodiversity, or to ask questions regarding the patterns of niche evolution (Araújo & Peterson, 2012).”

Caveats and assumptions are a pre-requisite in research using models and should not be discouraged from the study of science.

Therefore, more revisions, especially on using the terms 'refugia' in the expressions of the future distributions, are still needed. Although authors listed diverse definitions of refugia, they did not point out the common essential of these definitions: refugia is defined based on the contrasted distributions of one species (or one flora) at different stages because of the climatic oscillations.

The different definitions of refugia were already provided in the former version of the ms. and while the reviewer acknowledged this, the definition we used was not the one that they liked, so it was assumed to be wrong. In the current revision we further clarified our definition: *In the present study we adopt the prerequisite of long-term persistence as the main trait defining refugia and, for the context of relict plant species in East Asia, we define “long-term stable refugia” as the climatically suitable areas that allowed the persistence (in contrast to other areas) of ancient lineages during the Pleistocene climatic oscillations and that probably will do under a scenario of global warming. We have also introduced a clarification between brackets in the last sentence of the Introduction, as follows: Finally, we identify long-term (at least since the LGM to the year 2070) stable refugia maintaining ancient lineages and propose establishment of protected areas for the long-term stable refugia.*

The ongoing debate within the scientific community regarding which definition of refugia is more correct than another is far outside of the scope of this paper. Nevertheless, we clarified which of these definitions we are following and made sure that the term “long-term stable refugia” (and the more general term “climate refugia”) remains internally consistent throughout the ms.

SW China is considered as the refugia of these ancient lineages at LGM or the current, because before LGM and current, they had distinctly larger distributions. At the current or aft LGM with warmer climate, if distributions of these relic species are determined by climate, why did not they expand as numerous species? The limited distributions of these relic species may be mainly shaped by their limited evolutionary potentials (greatly reduced in dispersal and adaptation because of the too small population sizes of these endangered relic species) and human activities. Therefore, the term 'refugia' coined for climatic oscillations of different stages and modelling researches are really difficult to feel comfortable under such a scenario.

Evolutionary theory is pretty clear on this matter. Organisms can either 1) move, 2) adapt, or 3) die. Movement and adaptation require competition for space and resources and in Asia there are a lot of taxa competing for the same resources, so one would not expect to see wide expanses of a monoculture. The size of the population does contribute to the potential of future generations to evolve as the physical conditions around them continue to change; however, it is not required. *Wollemia* is a good example of an extremely small population of trees surviving millions of years of change in a very small area. In general and on the basis of modelling using exclusively climatic variables, there is a general loss of suitable core areas with the highest richness (51-128) and when all the species were accounted for, only a slight gain of suitable areas was attained (ca. 3% to 12%, see Supplementary Table 10). However, even assuming these expansions (e.g., NE China, North Korea, N Japan, or S Kamchatka), these will be unlikely given the dispersal limitations of the relict species and the unlikely conservation of these habitats by the year 2070, as clearly stated in lines 274-276 in our 05/31/18 ms). We agree with our Reviewer #2's opinion.

We and other researchers know that dispersal and adaptation could affect species distribution. However, as a matter of fact there is far less evidence for it and far less data on it than the great data set on species distribution over time and of climate data overtime that we and most other researchers prefer to use. In our ms. we use the climate modeling approach, which cannot be replaced by Reviewer #4's suggested approach of reduced dispersal and adaptation.

1. One similar research on plants in southwest China also used the similar niche modelling way (in fact more complex and better) to predict the distribution changes of plants in this region in response to the climate changes (Liang et al. 2018, Journal of Biogeography, DOI: 10.1111/jbi.13229). They pointed out many caveats in such modelling results.

As explained above, we have pointed out all potential caveats concerning modelling as far as possible. Niche modelling is just another approach to trying to predict how plants move in space and time. Despite the fact that the reviewer considers this to be a better

tool (and we must presume better than anything else) because it is more complex is just another opinion.

2. In the future, these ancient species still persist in south west China although shifting southward. As pointed out by authors 'the areas as a whole will probably expand. Therefore, based on these range changes as well as the climatic warming, it should be very careful to conclude about the 'permanent' refugia. Under this scenario, as responded by authors, even if refugia may have different definitions (however, always against the clear backgrounds, for example, distributions in LGM versus pre- or post-LGM), it is still difficult to reason that future distributions are 'refugia'. What is the background for refugia in the future? Are you assuming: these ancient should be distributed in the whole northern Hemisphere in the current or in 2070? Therefore I could not follow their logic conclusions.

The reviewer's version of the definition and use of refugia is grounded in the LGM and any deviation from this concept has little or no meaning as is the case here. As explained above, only a slight expansion of the climatically suitable areas compared to the present will be reached (ca. 3% to 12%, see Supplementary Table 10). As shown in Fig. 4, species will mostly remain in SW China, but there could be a little northwards expansion (not a southward shifting as stated by Reviewer #4). However, the calculations included in Supplementary Tables 10 and 11 (and based on the results presented in Figs. 3 & 4) are strictly based on the potential richness of the studied relict species in the past (LGM or Holocene), present, and future (year 2070), not the "long-term stable refugia".

Please be aware that the concept "long-term stable refugia" is applied to the sum of the individual species ENMs after getting the individual refugial areas (i.e., the individual species binary maps at the different time scenarios were intersected, with only the overlapping areas being retained and thus regarded as the species "refugium") and not simply to the sum of individual models without doing these individual intersections (see lines 665-675 in the version of our 05/31/18 ms). This is the way to ensure that the obtained "long-term stable" refugial areas are those places where individual species have remained since the LGM to the year 2070 (see also Supplementary Table 9 for an additional explanation). So, the darkest areas in Figs. 5D, 5E, 5I, and

5J are the places where a large number of individual relict species have persisted since the LGM to (probably) the year 2070. We are afraid that Reviewer #4 might not have understood this point in our interpretation of long-term stable refugia. As broadly explained in our first response to the Editor, for the present study we have chosen the definition of climate refugia provided by Tzedakis et al. (2013, *Trends Ecol. Evol.* 28, 696–704) and Birks (2015, *Biodiversity* 16, 196–212).

3. In the Liang et al. (2018) paper, they compared the range sizes of all sampled species in the current, LGM and 2050. I am wondering whether the claim 'the areas as a whole will probably expand' should include meta-analyses of range size changes of the sampled ancient species during different stages. This will give a clear over-view of the distributions of these ancient lineages from LGM, the Current and 2070. If the overall distributions are expanding as similar to those species examined by Liang et al. (2018), the conclusion 'stable refugia' is not good to express this pattern. If the overall distributions are shrinking from LGM to the Current to 2070, you can say that 'this region' is a 'refugia' against the current or LGM distributions. Overall, I feel so weird to say 'refugia' when without distinct evidence for contrasted distribution range sizes between different stages.

As explained above, our approximation to get the location of “long-term stable refugia” is based on the overlap of ranges of individual species, and then these are stacked (summed) to get those regions with high concentrations of relict species that persisted in the same locations; that is, it is like a sum of ‘individual hotspots’ to obtain ‘general hotspots’ for the relict flora of East Asia. Thus, our approximation is not exactly that of the paper of Liang et al. (2018), so the results of the two manuscripts are not directly comparable.

4. I feel that it is better not to say 'the survival of these relic species was shaped by climate in Asia'. It is very obvious that most of these relic species could grow very well in European gardens, for example, *Davidiana* and other species. In fact, well growth and normal reproductions of these relic species in Europe and North America raised further niche modelling questions on their conclusions and assumptions. At least, the limitations for such modelling and those existing facts should be mentioned.

A similar question was already answered in the first round of revisions of the manuscript. As we wrote then, although these species could grow well in European and North American gardens, the appearance of new populations of these species/lineages would be due more to entelechy than reality (unless man-made reintroductions were undertaken). Moreover, it should take into account that most individuals or stands of these relict species that currently grow in gardens are the result of seed germination and sapling growth under greenhouse controlled conditions, as germination and recruitment are generally very poor under natural conditions. In other words, the existence of cultivated individuals of a given species in gardens of a given (non-native) region do not automatically mean that this species will be capable of surviving in the nature by its own.

Nevertheless, we acknowledge the limitations of modelling, and such limitations have been included within the current revised manuscript (and the sentence mentioned by Reviewer #4 also modified as follows: “*the survival of these relict species was mainly shaped by climate in Asia*”).

5. Another question related to the above questions, is that most of these relict or endangered species are also widely cultivated in Asia because of their great values in conservation, re-forestry, researches and others. I did not find the details that how they differentiated the natural distributions and artificial cultivation. If you google, you will find so many sites where such one relict species (for example, Davidiana, Ginkgo, Metasequoia) was cultivated.

Yes, some of the species included in our analyses are widely cultivated in East Asia, including *Ginkgo biloba*, *Metasequoia glyptostroboides*, or *Glyptostrobus pensilis*. As stated in lines 570-571 in the 05/31/18 version of our ms., “*For species that favor human cultivation, we only recorded data on their natural occurrences as based on our own expertise and field work*”.

6. Another finding that rich relict flora occurred in the eastern Asia (isoline by precipitation) is similar to that recently reported in Nature (Lu et al. 2018, evolutionary history of the angiosperm flora of China). IN this nature paper, they also pointed out many relict lineages occurred in the eastern Asia. This work should be cited. They also mentioned that SW

China served as the refugia for numerous ancient lineages. However, their conclusions were based on the current limited distributions with the ancient wide distributions. This logic expression can be followed.

Thank you for the suggestion. We have cited this excellent study and compared their results to ours in the section “Present-day relict species richness patterns in East Asia”. We did not cite this paper in our former version of the ms. because Lu et al. paper was not published at the time of the submission.

Sincerely yours,

Cindy Q. Tang